# A FLEXIBLE GENERATIVE MODEL FOR HETEROGENEOUS TABULAR EHR WITH MISSING MODALITY

**Huan He**
Department of Biostatistics
University of Pennsylvania
huan.he@pennmedicine.upenn.edu

**William Hao** [*]
Department of Computer Science
Emory University
william.hao@emory.edu

**Yuanzhe Xi**
Department of Mathematics
Emory University
yuanzhe.xi@emory.edu

**Yong Chen**
Department of Biostatistics
University of Pennsylvania
ychen123@upenn.edu

**Bradley Malin**
Department of Biomedical Informatics
Vanderbilt University
b.malin@vumc.org

**Joyce C Ho**
Department of Biostatistics
Emory University
joyce.c.ho@emory.edu

## ABSTRACT

Realistic synthetic electronic health records (EHRs) can be leveraged to accelerate methodological developments for research purposes while mitigating privacy concerns associated with data sharing. However, the training of Generative Adversarial Networks remains challenging, often resulting in issues like mode collapse. While diffusion models have demonstrated progress in generating quality synthetic samples for tabular EHRs given ample denoising steps, their performance wanes when confronted with missing modalities in heterogeneous tabular EHRs data. For example, some EHRs contain solely static measurements, and some contain only contain temporal measurements, or a blend of both data types. To bridge this gap, we introduce FLEXGEN-EHR– a versatile diffusion model tailored for heterogeneous tabular EHRs, equipped with the capability of handling missing modalities in an integrative learning framework. We define an optimal transport module to align and accentuate the common feature space of heterogeneity of EHRs. We empirically show that our model consistently outperforms existing state-of-the-art synthetic EHR generation methods both in fidelity by up to 3.10% and utility by up to 7.16%. Additionally, we show that our method can be successfully used in privacy-sensitive settings, where the original patient-level data cannot be shared.

## 1 INTRODUCTION

The widespread digitization of health data has enabled the training of deep learning models for precision medicine, in the form of personalized prediction of risks and health trajectories (Rajkomar et al., 2018; Miotto et al., 2016). However, there are various concerns over patient privacy that need to be accounted for in order to collect the large quantities of data needed to train robust models. As such, it is challenging for researchers to obtain access to real electronic health records (EHRs). One approach to mitigate privacy risks is through the practice of de-identification in the form of data perturbation and randomization (El Emam et al., 2015; McLachlan et al., 2016). However, the blind application of de-identification leads to records that are vulnerable to re-identification attacks (Narayanan & Shmatikov, 2008). An alternative approach that is receiving increasing attention is the creation and dissemination of synthetic datasets that aim to capture many of the complexities of the

---

[*]Equal Contribution

original data set (e.g., distributions, non-linear relationships, and noise). Synthetic data can yield records that are robust against re-identification. To support the creation of realistic and synthetic data, generative models have emerged as the key element to advance precision medicine.

Although several distinguished efforts for synthetic EHR generation exist (Bing et al., 2022; Choi et al., 2017; Torfi & Fox, 2020; Yan et al., 2020; Li et al., 2021a; Theodorou et al., 2023), designing an effective generative model for EHRs remains a challenging task for two key reasons: (1) heterogeneous features encompassing both static and temporal measurements and (2) missing data. EHRs consist of diverse and multi-dimensional data that contain static features (e.g., race and gender) and temporal measurements (e.g., heart rate, temperature, blood pressure). We refer to this as heterogeneous tabular EHRs (i.e., not including unstructured text). Existing works (Choi et al., 2017; Torfi & Fox, 2020; He et al., 2023) extended classical deep generative models to EHRs, building upon generative adversarial networks (GANs), autoencoders, or diffusion models. However, these methods are limited to generating static measurements including billing codes, and ignore the temporal features (e.g., lab results repeated over time), hindering their utility for downstream tasks. On the other hand, (Biswal et al., 2021) focused on generating temporal features only. EHR-M-GAN (Li et al., 2021b) circumvents this problem by training two separate encoders for each modality (static and temporal features), but it lacks the capability of training using data with missing modality.

Missing data, is a complex and pervasive problem in medical records and public health research, affecting both randomized trials and observational studies (Haneuse et al., 2021). Nevertheless, existing methods assume completeness of the data, yielding less reliable generation when facing missing data. Reasons for missing data can vary substantially across studies because of dropout or loss to follow-up, missed study visits, or an unrecorded measurement during an office visit. Merely removing missing data is not feasible because the learned generative model is likely to suffer from selection bias, whereby the results downstream utility (ie, to the intended or target patient population) is compromised). This underscores the need for generative methods for EHRs that 1) handle heterogeneous (or mixed-type) EHRs and 2) expand the scope of existing generative methods by supporting missing observations of EHRs. A clear articulation of assumptions is critical because of the different mechanisms that can induce missing data. We consider the scenario of missing modality not at random (MMCAR) in work suggested by Tang et al. (2020); Haneuse et al. (2021); Gianfrancesco & Goldstein (2021). In this scenario, not every record will have data associated with each modality. For example, medications (documented as static data) may not have corresponding records of the measurements of patients physiological status due to loss of records. Another example is protecting patient privacy by omitting one modality where they can be easily identified (e.g., a woman above 100 with otherwise similar temporal features to the population).

**Present Work.** To address the aforementioned limitations, for the first time, we introduce FLEXGEN-EHR, a flexible generative framework for simultaneously synthesizing heterogeneous longitudinal EHR data. Specifically, we focus on generating both static and temporal records jointly. Patient trajectories with high-dimensionality and heterogeneous data types (both continuous-valued and discrete-valued timeseries) are generated while the underlying temporal dependencies and temporal-static correlations are captured.

In summary, our contributions are as follows: ① We formalize the challenge of generating heterogeneous Electronic Health Records (EHR) in the presence of missing modality.

② We present FLEXGEN-EHR, a latent diffusion method demonstrating superior generation fidelity, evidenced by a reduction of Maximum Mean Discrepancy (MMD) by up to 3.10%, and enhanced utility, with an increase in Area Under the Precision-Recall Curve (AUPR) by up to 7.16%.

③ We introduce an innovative solution to the missing modality issue by formulating an optimal transport problem in the embedding space, enabling the construction of meaningful and reasonable latent embedding pairs to solve the missing correspondence in the data.

④ We empirically verify that FLEXGEN-EHR maintains high standards of generation and utility even in instances of missing modality, solidifying its applicability and reliability in practical, real-world scenarios involving incomplete data.

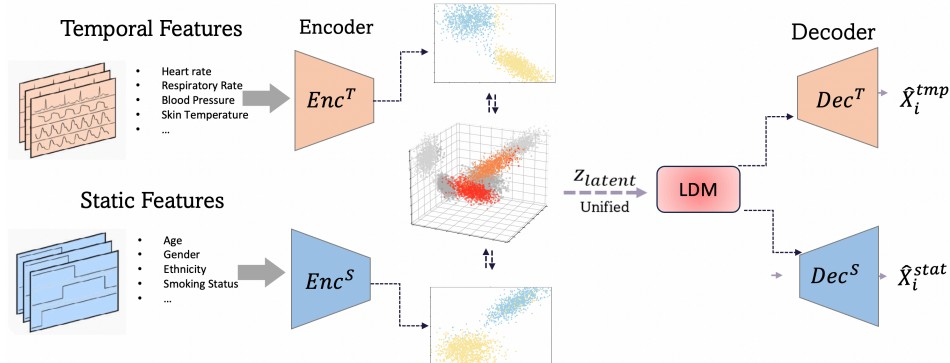

Figure 1: **Conceptual model illustration** FLEXGEN-EHR allows training using data with missing modality (details in 4.3).

## 2  RELATED WORK

Here, we discuss the related works on synthetic EHR generation and diffusion-based models.

In this context, we use 'categorical' and 'static' interchangeably, both referring to features that fall within defined categories and remain consistent. Similarly, 'numerical' and 'temporal' are used synonymously, denoting features that can vary over time.

**Synthetic EHR Generation.** The predominant approaches for EHR tabular generation currently rely on GAN-based methods, which utilize two neural networks: the generator and the discriminator. MedGAN (Choi et al., 2017), CorGAN(Torfi & Fox, 2020), and EHR-WGAN (Zhang et al., 2019) utilize GANs for generating patient feature matrices (diagnosis only). However, challenges still remain for GAN-based methods, such as the mode collapse problem (Li et al., 2021c) and the alignment between heterogeneous data types.

While several works focus on static data generation such as MedGAN and CorGAN, it's essential to note that real-world EHRs encompass a mixture of heterogeneous data types. These include temporal features (e.g., blood test results) and static features (e.g., sex and ethnicity), also denoted as numerical and categorical features. Although VAE performs the common method for static feature generation, when it comes to synthesizing temporal features, current GAN-based models, such as C-RNN-GAN (Mogren, 2016), TimeGAN (Yoon et al., 2019), rely heavily on adopting recurrent neural networks (RNNs) for both their generator and discriminator components. Therefore, in the context of heterogeneous Tabular EHR data generation, GAN-based methods face challenges in learning joint distribution representation due to the diverse structures of their generators.

Very recently, EHR-M-GAN (Li et al., 2021b) is proposed as a method for synthesizing heterogeneous EHR data. However, it operates under the assumption that representations learned by autoencoders can seamlessly integrate with those from bidirectional Long Short-Term Memory(BLSTM) (Zhou et al., 2016), a premise that might not be reliable given the inherent structural differences between the models. In contrast, FLEXGEN-EHR introduces a integrative learning framework. For both static and temporal features, they are learned under a separate VAE architecture, which naturally aligns common feature space of heterogeneity for EHR Tabular datas.

**Tabular Diffusion.** First proposed in (Sohl-Dickstein et al., 2015), diffusion models are a family of latent variable generative models characterized by a forward and a reverse Markov process. Diffusion models have excelled in various image-generation tasks and now extended beyond computer visions (Chen et al., 2021; Vahdat et al., 2021). However, only few works have been proposed to introducing diffusion models for tabular data generation so far, especially when encompassing with heterogenous datasets like EHRs.

As an early adopter of diffusion models for EHR generation, MedDiff (He et al., 2023) primarily focuses on utilizing Gaussian diffusion process to generate numerical EHR data. But it is important to note that real-world EHRs is formed by a mixture of numerical features (eg., respiration rate) and categorical features (eg., LOS and mortality). MedDiff generates diagnosis count matrix only, leav-

ing out numerical features unsolved. The more recent model, TabDDPM (Kotelnikov et al., 2023), addresses this problem by generating heterogeneous tabular data encompassing both static and temporal features, which could be denoted as numerical and categorical as well. It employs Gaussian quantile transformation for temporal features, and a one-hot encoder for each static features. A recent study (Ceritli et al., 2023) evaluates the performance of implementing TabDDPM into EHR generation, which out performs other contemporary models.

A common observation among existing diffusion-based EHR models is their inability to generate categorical features or their tendency to treat numerical and categorical features independently. Yet, in practical scenarios like bedside data analysis in hospitals, numerical features (eg., respiration rate) and categorical features (eg., diagnosis and admission type) often have intrinsic logical relationships. For a synthetic model to generate a realistic heterogeneous tabular EHR dataset, it must adeptly discern and represent the underlying relationships between numerical and categorical features. Different from TabDDP that concatenates the numerical and categorical features to the network, FLEXGEN-EHR adeptly discerns and represents the underlying relationships between static and temporal features.

## 3 PRELIMINARIES

**Diffusion.** Diffusion models leverage a pre-defined forward process in training, where a clean distribution $q(x_0)$ can be corrupted to a noisy distribution $q(x_t|x_0)$ at a specified timestep $t$. Given a pre-defined variance schedule $\{\beta_t\}_{1:T}$, the noisy distribution at any intermediate timestep is given by:

$$q(x_t|x_0) = \mathcal{N}\left(\sqrt{\bar{\alpha}_t}x_0, (1 - \bar{\alpha}_t)I\right); \quad \bar{\alpha}_t = \prod_{i=1}^{t}(1 - \beta_i).$$

To reverse such forward process, a generative model $\theta$ learns to estimate the analytical true posterior in order to recover $x_{t-1}$ from $x_t$ as follows:

$$\min_{\theta} D_{KL}[q(x_{t-1}|x_t, x_0)||p_\theta(x_{t-1}|x_t)]; \quad \forall t \in \{1, ..., T\},$$

and such an objective can be reduced to a simple denoising estimation loss []:

$$\mathcal{L}_{\text{DDPM}} = \mathbb{E}_{t, x_0 \sim q(x_0), \epsilon \sim \mathcal{N}(0, I)} \left[\|\epsilon - \epsilon_\theta(\sqrt{\bar{\alpha}}x_0 + \sqrt{1 - \bar{\alpha}_t}\epsilon, t)\|^2\right] \tag{1}$$

For the case where label information is available, the model is trained to estimate the noise as above in both conditional cases $\epsilon_\theta(x_t, y, t)$ with data-label pairs $(x_0, y)$ and unconditional case $\epsilon_\theta(x_t, t)$. In the sampling, the label-guided model estimates the noise with a linear interpolation

$$\hat{\epsilon} = (1 + \omega)\epsilon_\theta(x_t, y, t) - \omega\epsilon_\theta(x_t, t)$$

to recover $x_{t-1}$, which is often referred to as Classifier-Free Guidance (CFG) (Ho, 2022).

**Latent Diffusion Model.** To improve the efficiency, the Latent Diffusion Model (LDM) (Rombach et al., 2022b) introduces an explicit separation between the compressive and generative learning phases of training diffusion models. Central to this approach is the use of an autoencoding model, consisting of an encoder $E(\cdot)$ and a decoder $D(\cdot)$. This autoencoder is designed to capture a compressed latent space for diffusion model, which, upon decoding, closely resembles the original data space in its perceptual attributes.

Given a tabular data entry $x$ with dimensions $x \in \mathbb{R}^{H \times W}$, the encoder $E$ maps $x$ into a latent representation denoted by $z = E(x)$. Subsequently, the decoder $D$ reconstructs the original data entry from zz, resulting in $\tilde{x} = D(z) = D(E(x))$. This latent space, with dimensions $z \in \mathbb{R}^{h \times w}$, offers the advantage of reduced computational demands while preserving the perceptual integrity of the regenerated samples.

The objective function tailored for training the LDM is expressed as:

$$L(\theta) = \mathbb{E}_{z_t \sim q(z_t|z), z = E(x), t \sim [0,1]} \left[\omega_t \cdot \|F_\theta(z_t, t) - z\|_2^2\right] \tag{2}$$

Here, the latent representation $z$ is derived during the training process using Encoder $E$. Once generated, $z$ can be decoded back to its original data form using $D$.

### 3.1 Optimal Transport

Optimal transport formalizes the problem of finding a minimum cost mapping between two point sets, viewed as discrete distributions. Specifically, we assume two empirical distributions over embeddings, e.g., embeddings of static features $\mathbf{z}_i^{\mathcal{S}}$ and embeddings of temporal features $\mathbf{z}_i^{\mathcal{T}}$

$$\mu = \sum_{i=1}^{I} \mathbf{p}\mathbf{z}_i^{\mathcal{T}} \text{ and } \nu = \sum_{j=1}^{J} \mathbf{q}\mathbf{z}_j^{\mathcal{T}} \tag{3}$$

Here, $\mathbf{p}$ and $\mathbf{q}$ are non-negative vectors of length $I$ and $J$ that sum up to 1. We denote their probabilistic couplings, and cost matrix $\mathbf{C}$, as: We find a transportation map $\Gamma$ realizing :

$$\inf_{T} \left\{ \int_{\mathcal{X}} c(\mathbf{x}, \Gamma(\mathbf{x})) d\mu(\mathbf{x}) \mid \Gamma_{\#}\mu = \nu \right\}, \tag{4}$$

where the cost $c(\mathbf{x}, \Gamma(\mathbf{x}))$ is typically just $\|\mathbf{x} - \Gamma(\mathbf{x})\|$ and $\Gamma_{\#}\mu = \nu$ implies that the source points must exactly map to the targets.

## 4 FlexGen-EHR

### 4.1 Problem Formulation

**Heterogeneous Tabular EHR Generation.** Given heterogeneous tabular EHR data $\mathcal{D} = \left\{ \left( \mathbf{x}_i^{\mathcal{S}}, \mathbf{x}_i^{\mathcal{T}}, y_i \right) \right\}_{i=1}^{N}$ of $N$ electronic records where $\mathbf{x}_i^{\mathcal{T}} \in \mathbb{R}^m$ contains time-invariant features, $\mathbf{x}_i^{\mathcal{T}} \in \mathbb{R}^{T \times d}$ contains time-dependent features, and $y_i$ represents the label information of interest. Here, $N$ refers to the number of examples (unique IDs in the data table), $T$ is the number of timesteps after "discretizing" the observation period into time bins of size. The dimensionalities of the time-invariant and time-dependent features are denoted by $m$ and $d$, respectively. The goal is to generate synthetic EHR data $\hat{\mathcal{D}}$ such that $\mathcal{L}(\mathcal{D}, \hat{\mathcal{D}})$ is minimized where $\mathcal{L}$ is any divergence measurement, such as mean-squared loss and maximum mean discrepancy (MMD).

**Heterogeneous Tabular EHR Generation with missing modalities.** The above problem can be extended to the setting where not every record will have data associated with each modality. Thus, the heterogeneous tabular EHR data can take the form $\mathcal{D} = \left\{ \left( \mathbf{x}_i^{\mathcal{S}}, y_i \right), \left( \mathbf{x}_j^{\mathcal{T}}, y_j \right), \left( \mathbf{x}_k^{\mathcal{S}}, \mathbf{x}_k^{\mathcal{T}}, y_k \right) \right\}$ for $i = 1, \cdots, I, j = 1, \cdots J, k = 1, \cdots, K$. Here, $I, J, K$ represent the number of records containing only static, only temporal, and both types of information, respectively.

### 4.2 Latent Diffusion Model on EHRs

Given a sample $\left( \mathbf{x}_i^{\mathcal{S}}, \mathbf{x}_i^{\mathcal{T}}, y_i \right)$, FlexGen-EHR utilizes a dual encoder-decoder framework to independently obtain static and temporal latent embeddings, denoted as $z_i^{\mathcal{S}}$ and $z_i^{\mathcal{T}}$. Specifically, an encoder $Enc^{\mathcal{S}}$ operates on static features, embedding the patient information as $z_i^{\mathcal{S}} = Enc^{\mathcal{S}}(\mathbf{x}_i^{\mathcal{S}})$, while another encoder, $Enc^{\mathcal{T}}$, is specialized for temporal features, representing repeated measurements. The implementation of two distinct encoders is motivated by the fact that a single encoder is inadequate to capture meaningful embeddings for both feature types due to the distinctive domains of static and temporal features. Our empirical investigations confirm that utilizing an LSTM encoder for temporal features and an MLP encoder for static ones yields superior quality embeddings. Following the encoding phase, these separately constructed latent representations are concatenated to construct a fused latent representation, represented as $z_i = [z^{\mathcal{T}}; z^{\mathcal{S}}]$. A latent diffusion model $G$ is then trained on this unified representation $z_i$.

The choice of using a latent diffusion model is significant not only because generation with diffusion on a compact scale is more expedient, but also due to our observations that, given the high-dimension of EHR data ($d > 5000$), leveraging diffusion models or even the recently proposed TabDDPM tends to encounter training failures and generates samples at least 5 times slower. Thus the integration of a dual-encoder-decoder framework and a latent diffusion model is synergistic, culminating in the achievement of hyper-parameter robustness, high-quality synthetic samples, and more efficient generation procedures. This optimized combination ensures the precise and coherent embedding of

both static and temporal features, and provides a comprehensive and nuanced representation of the complex, multifaceted data encountered in healthcare settings.

## 4.3 LATENT SPACE ALIGNMENT

The above section assumes that both modalities are present, which cannot readily handle the missing modality. Here, we define the optimal transport problem for solving data with missing modalities such that $\mathbf{x}_i = \left(\mathbf{x}_i^{\mathcal{S}}, \text{NA}, y_i\right)$ or $\mathbf{x}_i = \left(\text{NA}, \mathbf{x}_i^{\mathcal{T}}, y_i\right)$, where NA denotes not available. Instead of mapping-based methods, which rely on nearest-neighbor computation to infer the NA data, we observed that latent space embedding models, trained on disparate features, manifested analogous geometric patterns and behaviors. This drives us to posit that the latent embedding spaces of heterogeneous EHRs can potentially be transformed reciprocally through linear transformations.

Specifically, suppose the sample size of $\mathbf{z}_i^{\mathcal{S}}$ and $\mathbf{z}_i^{\mathcal{T}}$ are identical, let $\mathbf{Z}^{\mathcal{T}} = \left[\mathbf{z}_1^{\mathcal{T}}, \ldots, \mathbf{z}_I^{\mathcal{T}}\right] \in \mathbb{R}^{lt \times I}$ and $\mathbf{Z}^{\mathcal{S}} = \left[\mathbf{z}_1^{\mathcal{S}}, \ldots, \mathbf{z}_J^{\mathcal{S}}\right] \in \mathbb{R}^{ls \times I}$ represent the embedding matrices of temporal and static features, respectively. Here, $lt$ and $ls$ denote the size of the embeddings and $I$ represents the number of samples. A viable solution entails solving the following linear system:

$$\min_{\mathbf{A} \in O(l)} \|\mathbf{Z}^{\mathcal{S}} - \mathbf{A}\mathbf{Z}^{\mathcal{T}}\|_F^2, \tag{5}$$

where $O(l) = \left\{\mathbf{A} \in \mathbb{R}^{ls \times lt} \mid \mathbf{A}^\top \mathbf{A} = \mathbf{I}\right\}$. This formulation has a closed-form solution that can be easily obtained. However, the linear system approach only solves the supervised version of the problem as it requires a known correspondence between the columns of $\mathbf{Z}^{\mathcal{S}}$ and $\mathbf{Z}^{\mathcal{T}}$. Clearly, it cannot solve the alignment problem without the correspondence caused by the missing modalities.

We solve this issue by "imputing" the correspondence via a modified Gromov-Wasserstein-based manifold alignment algorithm. The algorithm exploits the *inner* structure of the embedding space and uses the available label information as *global* correspondence between two spaces. We consider two measure spaces expressed in terms of within-embedding space similarity (cosine similarity used here) matrices $\mathbf{C}^{\mathcal{T}} \in \mathbb{R}^{I \times I}$ and $\mathbf{C}^{\mathcal{S}} \in \mathbb{R}^{J \times J}$. For clarity, we represent $i, j, k,$ and $l$ as indices for individual samples, acknowledging a slight abuse of notation herein. We now define a loss function between similarity pairs: $L : \mathbb{R} \times \mathbb{R} \to \mathbb{R}$, where $L\left(\mathbf{C}_{ik}^{\mathcal{T}}, \mathbf{C}_{jl}^{\mathcal{S}}\right)$ measures the discrepancy between the distances $\mathbf{C}_{ik}^{\mathcal{T}}$ and $\mathbf{C}_{jl}^{\mathcal{S}}$. In this work, we define $L\left(\mathbf{C}_{ik}^{\mathcal{T}}, \mathbf{C}_{jl}^{\mathcal{S}}\right) = \frac{1}{2}(y_i y_k \mathbf{C}_{ik}^{\mathcal{T}} - y_j y_l \mathbf{C}_{jl}^{\mathcal{S}})^2$. It can be understood as the cost of "matching" $i$ to $j$ and $k$ to $l$. In addition, it avoids misalignment by penalizing the mis-correspondence when pairs have similarities between the embedding space but have different outcomes. All the relevant values of $L(\cdot, \cdot)$ can be put in a 4-th order tensor $\mathbf{L} \in \mathbb{R}^{I \times I \times J \times J}$, where $\mathbf{L}_{ijkl} = L\left(\mathbf{C}_{ik}^{\mathcal{T}}, \mathbf{C}_{jl}^{\mathcal{S}}\right)$. Similar to the definition 3.1, we seek a coupling $\Gamma \in \mathbb{R}^{I \times J}$ specifying how much mass to transfer between each pair of points from the two embedding spaces. The Gromov-Wasserstein problem is then defined as solving

$$\text{GW}\left(\mathbf{C}^{\mathcal{T}}, \mathbf{C}^{\mathcal{S}}, \mathbf{p}, \mathbf{q}\right) = \min_{\Gamma \in \Pi(\mathbf{p}, \mathbf{q})} \sum_{i,j,k,l} \mathbf{L}_{ijkl} \Gamma_{ij} \Gamma_{kl} - \epsilon H(\Gamma), \tag{6}$$

where $\Gamma_{ij}$ is the relative probability that matches embedding $\mathbf{z}_i^{\mathcal{T}}$ to $\mathbf{z}_j^{\mathcal{S}}$. We add an entropic regularization penalty to solve the problem faster. Once we have solved equation 6, the optimal transport coupling $\Gamma$ provides an explicit (soft) matching between temporal and static features, which can be interpreted as a probabilistic translation: for every pair of embeddings, it provides a likelihood that these two embeddings are correspondent of each other. Now, we can solve the least square problem with full correspondence information:

$$\min_{\mathbf{A}} \|\mathbf{Z}^{\mathcal{S}} \Gamma - \mathbf{A}\mathbf{Z}^{\mathcal{T}}\|_F^2 \tag{7}$$

Instead of equation 5, this formulation can achieve alignment when the number of samples with missing modalities does not necessarily to be equal to that of non-missing data becasue the learning problem is not a supervised correspondence problem. Suppose static features in $\mathbf{x}_i$ are not available, we can obtain the most likely latent embedding $\mathbf{z}_i^{\mathcal{S}}$ by running a simple transformation $\mathbf{A}_{:i}\mathbf{z}_i^{\mathcal{T}}\Gamma_{i:}^{-1}$. To obtain $\mathbf{z}_i^{\mathcal{T}}$ from $\mathbf{z}_i^{\mathcal{S}}$, resolving the problem considering $\mathbf{z}_i^{\mathcal{S}}$ as the source and $\mathbf{z}_i^{\mathcal{T}}$ as the target is necessary to obtain a reliable and accurate solution.

**Generation.** Next, we elucidate the procedure for generation. Whether a modality is missing or present, we demonstrate that FLEXGEN-EHR can train a latent diffusion model using fused representations $z_i = [z^{\mathcal{T}}; z^{\mathcal{S}}]$, as shown in equation 2. If there is missing data, FLEXGEN-EHRcan "impute" the full representations by solve the OT problem equation 6 to find reasonable corresponding features.

Upon the successful training of the diffusion model, our first step is to generate synthetic fused representations. Subsequently, decoupled embeddings (by simply splitting) into respective decoders as follows:

$$\hat{\mathbf{x}}_i^{\mathcal{T}} = Dec^{\mathcal{T}}(z^{\mathcal{T}}), \quad \hat{\mathbf{x}}_i^{\mathcal{S}} = Dec^{\mathcal{S}}(z^{\mathcal{S}})$$

The overview of FLEXGEN-EHRis presented in Figure 1.

## 5 EXPERIMENTS

### 5.1 EXPERIMENTAL SETUP

**Baselines.** In our study, we consider six methods as baselines: i) VAE (Kingma & Welling, 2022), a Variational Autoencoder model, traditionally used in generating high-dimensional data; ii) MEDGAN (Choi et al., 2017), a GAN-based model that generates low-dimensional synthetic records and decoded with an autoencoder; iii) CORGAN (Torfi & Fox, 2020), another GAN-based model that, akin to MedGAN, amalgamates Convolutional Generative Adversarial Networks (ConvGANs) and Convolutional Autoencoders to synthesize and reconstruct medical records; iv) EHR-M-GAN (Li et al., 2021b), a GAN-based model tailored for longitudinal heterogeneous EHRs; v) TABDDPM (Kotelnikov et al., 2023), a diffusion model specialized in handling tabular data with the unique capability of addressing the challenges presented by the heterogeneous nature of EHRs; and vi) LDM (Rombach et al., 2022a), a latent diffusion model that decomposes the generation process into a sequence of autoencoders and diffusion models (DMs).

These models introduce unique methodologies and focus on varying aspects of EHR data synthesis and reconstruction, providing a comprehensive perspective on the potential approaches and their efficacy in handling heterogeneous medical records. More details are available in Appendix B.1.

**Datasets.** We use two real-world de-identified EHR datasets, MIMIC-III (Johnson et al., 2016) and eICU (Pollard et al., 2018). Both are inpatient datasets that consist of varying lengths of sequences and include multiple static and temporal features with missing components.

We adopted the preprocessed datasets from FIDDLE (Tang et al., 2020). The cohort numbers and dimensionalities of extracted features are summarized in Table 1, where $m$ and $d$ denote the dimensions of static and temporal features respectively. More details on datasets and preprocessing are available at Appendix B.2.

Table 1: Summarization of the sample size and dimensionalities of extracted features. ARF means acute respiratory failure.

| MIMIC-III | **N** | m | d | eICU | **N** | m | d |
|---|---|---|---|---|---|---|---|
| In-hospital mortality (48 h) | 8,577 | 96 | 7,307 | In-hospital mortality (48 h) | 77,066 | 146 | 2,382 |
| ARF (4h) | 15,873 | 98 | 4,045 | ARF (4h) | 138,840 | 717 | 5,854 |

**Evaluation.** We evaluate the effectiveness of FLEXGEN-EHR on fidelity, utility on downstream machine learning tasks, and also demonstrate its ability to preserve privacy. *Fidelity:* We evaluate the quality of synthetic data through various metrics that assess how closely the synthetic data resembles real data. Following Li et al. (2021b); Yoon et al. (2023), we report $R^2$ values (the higher the better) and maximum mean discrepancy (the lower the better). The details can be found in B.3. *Utility:* We focus on prediction tasks and train a random forest (RF). We report the area under the ROC and PR curves (AUROC and AUPR, respectively) using test datasets. *Privacy:* Unlike de-identified data, there is no straightforward one-to-one mapping between real and synthetic data (generated from random vectors). However, there may be some indirect privacy leakage risks built on correlations between the synthetic data and partial information from real data. Following Torfi & Fox (2020);

Theodorou et al. (2023), we consider the membership inference attack that adversaries may apply to de-anonymize private data.

**Implementation Details.**

We perform experiments on two settings: i) an easier setting where we train models on data without missing modality, and ii) a harder setting where we randomly delete $p\%$ static samples and $q\%$ temporal features. We implemented FLEXGEN-EHR with PyTorch. For training the models, we used Adam (Kingma & Ba, 2015) with the learning rate set to 0.001, and a mini-batch of 128 on a machine equipped with one Nvidia GeForce RTX 3090 and CUDA 11.2. Hyperparamters of FLEXGEN-EHR are selected after grid search. We use a timestep of 50 and a noise scheduling $\beta$ from $1 \times 10^{-4}$ to $1 \times 10^{-2}$.

## 5.2 Q1: RESULTS-COMPARISON IN FIDELITY

We first evaluates the statistical similarity of the generated and real data. For each method, we generate a synthetic dataset of the same size as the training dataset. We calculate the probabilities for each feature (dimension-wise) within the real and synthetic datasets and then compute $R^2$ and MMD values. Results are presented in Table 2. Across all methods, we find that FLEXGEN-EHR consistently outperforms baselines by making an average improvement of $R^2$ (0.025) and MMD (0.031). In terms of correlation, FLEXGEN-EHR improves $R^2$ by 0.027, 0.019, 0.019, and 0.032 over the strongest baseline on each dataset respectively. In terms of distance, FLEXGEN-EHR reduces MMD by 0.030, 0.028, 0.014, and 0.049 over the strongest baseline on each dataset respectively. Although it is not our main contribution, results demonstrated that the improvement can be attributed to the dual encoder-decoder with latent diffusion.

Table 2: **Generation Fidelity.** $R^2$ (↑) correlation and MMD (↓) distance between synthetic and real datasets. FLEXGEN-EHRachieves the highest correlation.

| Dataset | MIMIC-Mortality | | MIMIC-ARF | | eICU-Mortality | | eICU-ARF | |
|---|---|---|---|---|---|---|---|---|
| | $R^2$ (↑) | MMD (↓) | $R^2$ | MMD | $R^2$ | MMD | $R^2$ | MMD |
| VAE | 0.632 ±0.052 | 1.514 ±0.048 | 0.577 ±0.074 | 0.981 ±0.048 | 0.694 ±0.031 | 0.857 ±0.029 | 0.655 ±0.034 | 0.802 ±0.016 |
| MEDGAN | 0.639 ±0.047 | 1.108 ±0.037 | 0.595 ±0.069 | 0.831 ±0.052 | 0.712 ±0.037 | 0.832 ±0.044 | 0.662 ±0.044 | 0.786 ±0.019 |
| CORGAN | 0.656 ±0.051 | 1.093 ±0.041 | 0.612 ±0.055 | 0.830 ±0.059 | 0.724 ±0.054 | 0.712 ±0.037 | 0.671 ±0.031 | 0.760 ±0.015 |
| TABDDPM | 0.395 ±0.060 | 1.636 ±0.067 | 0.525 ±0.059 | 1.182 ±0.041 | 0.653 ±0.046 | 0.774 ±0.045 | 0.617 ±0.038 | 0.784 ±0.012 |
| LDM | 0.694 ±0.038 | 0.755 ±0.042 | 0.658 ±0.052 | 0.635 ±0.039 | 0.758 ±0.049 | 0.662 ±0.038 | 0.736 ±0.033 | 0.710 ±0.017 |
| EHR-M-GAN | 0.712 ±0.044 | 0.711 ±0.045 | 0.676 ±0.047 | 0.614 ±0.043 | 0.781 ±0.042 | 0.617 ±0.033 | 0.762 ±0.019 | 0.667 ±0.020 |
| **FLEXGEN-EHR** | **0.739 ±0.045** | **0.681 ±0.042** | **0.695 ±0.051** | **0.586 ±0.045** | **0.800 ±0.037** | **0.603 ±0.035** | **0.794 ±0.024** | **0.618 ±0.019** |

## 5.3 Q2: RESULTS-COMPARISON IN UTILITY

Following the experimental setup in (Tang et al., 2020), we trained ML models to predict inhospital mortality and acute respiratory failure (ARF) in our evaluation of utility. Overall, FLEXGEN-EHR has won 4 out of 4 tests (2 metrics in 4 datasets) and makes an average improvement of AUROC (6.37%) and AUPR (7.16%) over with the strongest baseline across datasets.

Table 3: **Generation Utility.** The downstream classifier is trained using either real data (the first row) or synthetic data (2-8 rows). We report AUROC (↑) and AUPR (↑) evaluated on real test set. FLEXGEN-EHRachieves the highest comparable performance with real data.

| Dataset | MIMIC-Mortality | | MIMIC-ARF | | eICU-Mortality | | eICU-ARF | |
|---|---|---|---|---|---|---|---|---|
| | AUROC | AUPR | AUROC | AUPR | AUROC | AUPR | AUROC | AUPR |
| Real data (goal) | 0.856 ±0.032 | 0.445 ±0.117 | 0.817 ±0.023 | 0.657 ±0.029 | 0.841 ±0.012 | 0.401 ±0.027 | 0.714 ±0.014 | 0.269 ±0.032 |
| VAE | 0.731 ±0.031 | 0.394 ±0.102 | 0.693 ±0.019 | 0.582 ±0.025 | 0.732 ±0.012 | 0.298 ±0.029 | 0.634 ±0.012 | 0.188 ±0.035 |
| MEDGAN | 0.754 ±0.030 | 0.401 ±0.109 | 0.766 ±0.021 | 0.602 ±0.021 | 0.743 ±0.015 | 0.310 ±0.029 | 0.647 ±0.015 | 0.211 ±0.022 |
| CORGAN | 0.759 ±0.027 | 0.400 ±0.114 | 0.783 ±0.025 | 0.613 ±0.023 | 0.752 ±0.008 | 0.317 ±0.024 | 0.649 ±0.017 | 0.214 ±0.025 |
| TABDDPM | 0.712 ±0.028 | 0.375 ±0.135 | 0.788±0.024 | 0.626 ±0.027 | 0.741 ±0.012 | 0.305 ±0.027 | 0.615 ±0.015 | 0.205 ±0.028 |
| LDM | 0.772 ±0.036 | 0.408 ±0.122 | 0.775±0.020 | 0.621 ±0.022 | 0.749 ±0.011 | 0.322 ±0.025 | 0.668 ±0.019 | 0.223±0.029 |
| EHR-M-GAN | 0.778 ±0.035 | 0.409 ±0.109 | 0.787±0.025 | 0.645 ±0.024 | 0.753 ±0.010 | 0.326 ±0.028 | 0.681 ±0.021 | 0.216 ±0.031 |
| **FLEXGEN-EHR** | **0.794 ±0.035** | **0.428 ±0.116** | **0.792 ±0.017** | **0.674 ±0.025** | **0.764 ±0.013** | **0.356 ±0.028** | **0.689 ±0.014** | **0.254 ±0.026** |

Table 4: **Generation Privacy.** Accuracy is reported as the membership inference metrics.

| Dataset | MIMIC-Mortality | MIMIC-ARF | eICU-Mortality | eICU-ARF |
|---|---|---|---|---|
| Real (Ideal) | 0.5 | 0.5 | 0.5 | 0.5 |
| VAE | 0.512 | 0.488 | 0.519 | 0.487 |
| MEDGAN | 0.485 | 0.490 | 0.482 | 0.517 |
| CORGAN | 0.513 | 0.481 | 0.515 | 0.483 |
| TABDDPM | 0.483 | 0.521 | 0.521 | 0.481 |
| LDM | 0.482 | 0.518 | 0.519 | 0.483 |
| EHR-M-GAN | 0.485 | 0.514 | 0.517 | 0.482 |
| **FLEXGEN-EHR** | 0.483 | 0.484 | 0.519 | 0.481 |

## 5.4 Q3: RESULTS-COMPARISON IN PRIVACY

In this section, we quantify the vulnerability of all methods to adversarys membership inference attacks (Hayes et al., 2018). As shown in Table 4, there is not much difference in the privacy metrics (e.g., the accuracy of an attacker changed from 0.482 to 0.513 on MIMIC-Mortality). It shows that FLEXGEN-EHR maintains a level of privacy preservation that aligns with existing methods, while achieving superior data fidelity. It is essential to recognize that this trade-off between generation quality and privacy guarantee is a shared characteristic among all diffusion-based generative models.

## 5.5 Q4: RESULTS-HETEROGENEOUS TABULAR EHR GENERATION WITH MISSING MODALITY

We investigate the capability of FLEXGEN-EHR to generate with missing modality effectively. We randomly designate $p\%$ of samples as lacking temporal features and $q\%$ samples as lacking temporal features. We compare with EHR-M-GAN, the strongest baseline selected for this study. In the case of the baseline, we impute samples with absent modality using kNN and proceed to train both the generative and classification models, as EHR-M-GAN is not able to handle such scenarios. **Investigation into Generation Fidelity.** We report the fidelity metrics ($R^2$) in Figure 2 and 3. It can be observed that FLEXGEN-EHR maintains high generation fidelity even when number of samples with missing modality increases. **Examination of Generation Utility.** Subsequently, we assess the generation utility. The outcomes shown in Figure 4 demonstrate that the simple imputation of samples notably diminishes discriminative capability, whereas FLEXGEN-EHR can alleviate this problem, delivering performance that is comparable with full samples.

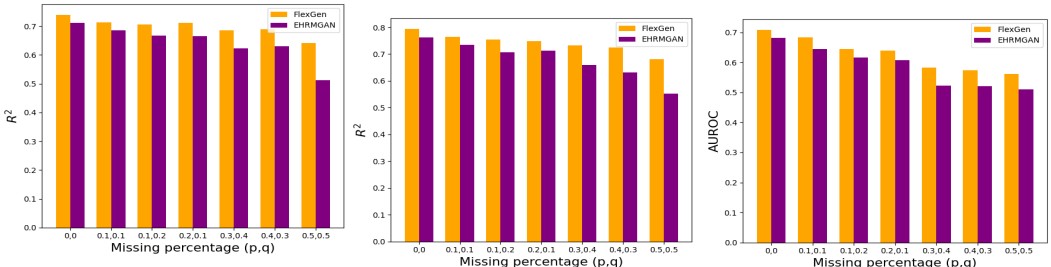

Figure 2: $R^2$ on MIMIC-Mortality

Figure 3: $R^2$ on eICU-ARF

Figure 4: AUROC on eICU-ARF

## 6 CONCLUSION

We introduce FLEXGEN-EHR, a novel generative model that is both flexible and easy-to-use, and can be trained on data with missing modality. We introduce an innovative solution to the missing modality issue by formulating an optimal transport problem in the embedding space, enabling the construction of meaningful and reasonable latent embedding pairs to solve the missing correspondence in the data. By constructing reasonable fused representations for data with missing modality, FLEXGEN-EHR is able to training using all samples without requiring removing then or other imputation techniques. Results demonstrate the potential of FLEXGEN-EHR as a general strategy for EHR generative models.

ACKNOWLEDGMENTS

This work was funded in part by the National Science Foundation (NSF) awards IIS-2145411, CNS-2124104, DMS-2208412, and National Institutes of Health (1R01LM014344, 1R01AG077820, R01LM012607, R01AI130460, R01AG073435, R56AG074604, R01LM013519, R56AG069880, U01TR003709, RF1AG077820, R21AI167418, R21EY034179)

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

## A    FURTHER DETAILS

### A.1    DIFFUSION FORWARD PROCESS

Given a data distribution $\mathbf{x}0q(\mathbf{x}0)\mathbf{x}_0 \sim q(\mathbf{x}_0)$, diffusion models gradually add noise to the original data distribution until it loses all the original information and becomes an entirely noisy distribution (as shown by the $xTx_T$ sample in Figure 1). DDPM convolves $q(\mathbf{x}_0)$ with an isotropic Gaussian noise $\mathcal{N}(0, \sigma_i^2\mathbf{I})$ in T steps to produce a noise corrupted sequence $\mathbf{x}_1, \mathbf{x}_2, \ldots, \mathbf{x}_T$. $\mathbf{x}_T$ will converge to an isotropic Gaussian distribution as $T \to \infty$. However, for DDIM, a different variance schedule is used to produce $\mathbf{x}_1, \mathbf{x}_2, \ldots, \mathbf{x}_T$. The explicit input distribution $q$ of DDIM is derived as :

$$q(\mathbf{x}_{1:T}|\mathbf{x}_0) := q(\mathbf{x}_T|\mathbf{x}_0) \prod_{t=2}^{T} q(\mathbf{x}_{t-1}|\mathbf{x}_t, \mathbf{x}_0) \tag{8}$$

where $q(\mathbf{x}_T|\mathbf{x}_0) = \mathcal{N}(\sqrt{\bar{\alpha}}\mathbf{x}_0, (1 - \bar{\alpha}_T)\mathbf{I})$ and for all $t > 1$,

$$q(\mathbf{x}_{t-1}|\mathbf{x}_t, \mathbf{x}_0) = \mathcal{N}(\sqrt{\bar{\alpha}_{t-1}}\mathbf{x}_0 + \sqrt{1 - \bar{\alpha}_{t-1} - \sigma^2}\frac{\mathbf{x}_t - \sqrt{\bar{\alpha}_t}\mathbf{x}_0}{\sqrt{1 - \bar{\alpha}_t}}, \sigma^2\mathbf{I})$$

Here $\bar{\alpha}$ controls the scale of noise added at each time step. At the beginning, noise should be small so that it is possible for the model to learn well, e.g., $\bar{\alpha}_1 > \bar{\alpha}_2 \cdots > \bar{\alpha}_T$.

The DDIM distribution is parameterized to guarantee the marginal density is equivalent to DDPM. However, the key difference between DDPM and DDIM is that the forward process of DDIM is no longer a Markov process. This allows acceleration of the generative process as multiple steps can be taken. Moreover, different reverse samplers can be utilized by changing the variance of the reverse noise. This means DDIM can be compatible with other samplers.

### A.2    DIFFUSION BACKWARD PROCESS

For DDPM, the forward process is defined by a Markov chain, and thus the true sample can be recreated from Gaussian noise $\mathbf{x}_T \sim \mathcal{N}(\mathbf{0}, \mathbf{I})$ by reversing the forward process. However, as noted above, DDIM relies on a family of non-Markovian processes. Denote $p_\theta$ as a parameterized neural network, the reverse process with a prior $p_\theta(\mathbf{x}_T) = \mathcal{N}(\mathbf{0}, \mathbf{I})$ can be computed as

$$p_\theta(\mathbf{x}_0|\mathbf{x}_1) = \mathcal{N}\left(\frac{\mathbf{x}_1 - \sqrt{1 - \bar{\alpha}_1}_\theta(\mathbf{x}_1, 1)}{\sqrt{\bar{\alpha}_1}}, \sigma_1^2\mathbf{I}\right), \quad p_\theta(\mathbf{x}_{t-1}|\mathbf{x}_t) = q(\mathbf{x}_{t-1}|\mathbf{x}_t, \frac{\mathbf{x}_t - \sqrt{1 - \bar{\alpha}_t}_\theta(\mathbf{x}_t, t)}{\sqrt{\bar{\alpha}_t}})$$

$$\tag{9}$$

### A.3    DIFFUSION CONDITIONED SAMPLING

To incorporate guidance in the absence of an independent classifier $f_\phi$, one could leverage the scores from both conditional and unconditional diffusion models as suggested . Specifically, the unconditional denoising diffusion model, represented as $p_\theta(\mathbf{x})$, is parameterized through a score estimator $\boldsymbol{\epsilon}_\theta(\mathbf{x}_t, t)$. On the other hand, the conditional model $p_\theta(\mathbf{x}|y)$ utilizes a score estimator $\boldsymbol{\epsilon}_\theta(\mathbf{x}_t, t, y)$. These two models can be learned through a single neural network.

To elucidate, a conditional diffusion model $p_\theta(\mathbf{x}|y)$ is trained on corresponding data sets $(\mathbf{x}, y)$. Through the training process, the conditioning variable $y$ is discarded intermittently at random intervals. This procedure ensures the model's capability to generate images without explicit conditioning, as expressed by the relation: $\boldsymbol{\epsilon}_\theta(\mathbf{x}_t, t) = \boldsymbol{\epsilon}_\theta(\mathbf{x}_t, t, y = \varnothing)$.

Furthermore, the gradient of an implicit classifier can be expressed utilizing both conditional and unconditional score estimators. Once incorporated into the classifier-guided modified score, there's no reliance on an independent classifier, as shown in the equations below:

$$\nabla_{\mathbf{x}_t} \log p\left(y \mid \mathbf{x}_t\right) = \nabla_{\mathbf{x}_t} \log p\left(\mathbf{x}_t \mid y\right) - \nabla_{\mathbf{x}_t} \log p\left(\mathbf{x}_t\right)$$

$$= -\frac{1}{\sqrt{1-\bar{\alpha}_t}}\left(\epsilon_\theta\left(\mathbf{x}_t, t, y\right) - \epsilon_\theta\left(\mathbf{x}_t, t\right)\right)$$

$$\bar{\epsilon}_\theta\left(\mathbf{x}_t, t, y\right) = \epsilon_\theta\left(\mathbf{x}_t, t, y\right) - \sqrt{1-\bar{\alpha}_t} w \nabla_{\mathbf{x}_t} \log p\left(y \mid \mathbf{x}_t\right) \tag{10}$$

$$= \epsilon_\theta\left(\mathbf{x}_t, t, y\right) + w\left(\epsilon_\theta\left(\mathbf{x}_t, t, y\right) - \epsilon_\theta\left(\mathbf{x}_t, t\right)\right)$$

$$= (w+1)\epsilon_\theta\left(\mathbf{x}_t, t, y\right) - w\epsilon_\theta\left(\mathbf{x}_t, t\right)$$

The parameter $w$ emerges as a weighting coefficient in the equation. Its role is to modulate the difference between the conditional and unconditional score estimators, effectively controlling the balance between the two. In essence, $w$ determines the extent to which the conditional guidance is emphasized over its unconditional counterpart.

### A.4 Algorithm of FlexGen-EHR

---

**Algorithm 1:** Training of FlexGen-EHR

---

**Input**: $\mathcal{D} = \left\{\left(\mathbf{x}_i^{\mathcal{S}}, y_i\right), \left(\mathbf{x}_j^{\mathcal{T}}, y_j\right), \left(\mathbf{x}_k^{\mathcal{S}}, \mathbf{x}_k^{\mathcal{T}}, y_k\right)\right\}$ for $i = 1, \cdots, I, j = 1, \cdots J, k = 1, \cdots, K$

A static encoder $Enc^{\mathcal{S}}$, a temporal encoder, $Enc^{\mathcal{T}}$

A static decoder $Dec^{\mathcal{S}}$, a temporal decoder, $Dec^{\mathcal{T}}$

A latent diffusion model $G$

**for** $e \leftarrow 1$ **to** $E$ **do**

    Compute static and temporal latent embeddings $z_i^{\mathcal{S}} = Enc^{\mathcal{S}}(\mathbf{x}_i^{\mathcal{S}}), z_i^{\mathcal{T}} = Enc^{\mathcal{T}}(\mathbf{x}_i^{\mathcal{S}})$

    Learn correspondece information $\Gamma$ by solving equation 6

    Learn $\mathbf{A}$ by solving $\min_{\mathbf{A}} \|\mathbf{Z}^{\mathcal{S}}\Gamma - \mathbf{A}\mathbf{Z}^{\mathcal{T}}\|_F^2$

    Impute missing embeddings by $\mathbf{z}_i^{\mathcal{S}} = \mathbf{A}_{:i}\mathbf{z}_i^{\mathcal{T}}\Gamma_{i:}^{-1}$

    Train $G$ using fused latent embeddings $z_i = [z^{\mathcal{T}}; z^{\mathcal{S}}]$

    Train encoders and decoders based on $G(z_i)$

**end**

Return $Enc^{\mathcal{S}}, Enc^{\mathcal{T}}, Dec^{\mathcal{S}}, Dec^{\mathcal{T}}, G$

---

## B Experimental Details

### B.1 Implmentations

**Encoder Selection.** we carefully selected the appropriate encoder. This consideration was applied to all our comparisons. For the extraction of temporal features, we utilized a 1d neural network (CNN) as the encoder. This configuration was kept consistent across all datasets to ensure a fair comparison, where differences in prediction performance could be attributed to the generative algorithm itself. The implementation of the CNN architecture was adapted from a recently published benchmark codebase in the literature Ragab et al. (2022). The 1D-CNN architecture consists of three blocks, each consisting of a 1D convolutional layer, followed by a 1D batch normalization layer, a rectified linear unit (ReLU) function for non-linearity, and finally, a 1D max-pooling layer. **VAE:.** It's a generative model that learns to encode and decode data in an unsupervised manner. The model is constructed with an encoder, a decoder, and a loss function designed to reconstruct data while enforcing a constraint on the encoding space. The encoder converts input data into a set of parameters in a latent space, typically representing the mean and variance of a probability distribution. These parameters are then sampled to generate new data points in the latent space, which the decoder reconstructs back into the original data space. **MEDGAN.** It's a discrete-valued EHR generative model. We flatten the input and use a two layer MLP to train the model. The first layer and second layer have a hidden size of 512 and 128 respectively. **CorGAN.** Similar to MedGAN, it is proposed to generate discrete EHR matrices. For all other baselines, we directly adopt implemen-

Table 5: Summary of MIMIC-III tables used in our analysis

| Table name | Description | Example variables |
|---|---|---|
| PATIENTS | Information on unique patients | Age, Sex |
| ADMISSIONS | Information on unique hospitalizations | Admission type, Admission location |
| ICUSTAYS | Information on unique ICU stays | Care unit, Ward ID, Admission-to-ICU time |
| CHARTEVENTS | Charted data, including vital signs, and other information relevant to patients care | Heart rate, Pain location, Daily weight |
| LABEVENTS | Laboratory test results from the hospital database | Lactate, WBC |
| INPUTEVENTS_MV | Fluid intake administered, including dosage and route (e.g., oral or intravenous) | NaCl 0.45%, Whole blood |
| OUTPUTEVENTS | Fluid output during the ICU stay OR urine | Stool |
| PROCEDUREEVENT_MV | Patients procedures during the ICU stay | CT scan, X-ray |
| MICROBIOLOGYEVENTS | Microbiology specimen from hospital database | Sputum |
| DATETIMEEVENTS | Documentation of dates and times of certain events | Last dialysis, Pregnancy due |

tations from the corresponding official websites and only modify necessary hyperparameters. Codes of baseline models are available online [1] [2] [3] [4] [5].

## B.2 DATASETS

MIMIC-III is an extensive database that houses anonymized data related to around sixty thousand critical care unit admissions from Beth Israel Deaconess Medical Center, gathered between 2001 and 2012. eICU is a large-scale, multi-center collection of anonymized health-related data, encompassing over 200,000 admissions to intensive care units throughout the United States from 2014-2015. We adopted the preprocessed datasets from FIDDLE (Tang et al., 2020). Table 5 and 6 are directly reused from (Tang et al., 2020) here. In table 5, files 'PATIENTS, ADMISSIONS, ICUSTAYS' correspond to static features and others are temporal features. In table 6, files 'patient, treatment' correspond to static features and others are temporal features

## B.3 EVALUATION

In our model, $R^2$ serves as an indicator of how well the synthetic data generated by FLEXGEN-EHR corresponds to the real-world data it aims to replicate. This metric is particularly relevant for evaluating the model's performance in terms of data fidelity. In contrast to $R^2$, MMD/KS-statics are both non-parametric tests used for determining if two samples are drawn from different distributions. By quantifying the maximum distance between these functions, MMD/KS-Statistics allow us to rigorously test the hypothesis that the synthetic and original data come from the same distribution. **Fidelity Metric: Maximum Mean Discrepancy (MMD).** Maximum Mean Discrepancy (MMD) is a statistical test used to measure the difference between two probability distributions. For two

---

[1] Code available at https://github.com/CompVis/latent-diffusion

[2] Code available at https://github.com/yandex-research/tab-ddpm

[3] Code available at https://github.com/mp2893/medgan

[4] Code available at https://github.com/astorfi/cor-gan

[5] Code available at https://github.com/jli0117/ehrMGAN

Table 6: Summary of eICU tables used in our analysis

| heightTable name | Description | Example variables |
|---|---|---|
| patient | Information on unique patients, hospitalizations, and ICU stays | Age, Sex
Hospital/ward ID |
| vitalPeriodic
vitalAperiodic | Vital signs measured through bedside monitors or invasively | Temperature
End Tidal CO2 |
| lab
customLab | Laboratory tests | CPK
troponin - I |
| medication
infusionDrug
intakeOutput | Active medication orders, the intake of drug through infusions, and intake/output of fluids | Morphine dosage
Dialysis total |
| microLab | Microbiology cultures taken from patients | Culture site (wound)
Organism |
| note
nurseAssessment
nurseCare
nurseCharting | Documentation of physician/nurse assessment | Abdominal pain
Psychological status
Respiratory rate |
| pastHistory | Relevant past medical history | Transplant
AIDS |
| physicalExam | Results of physical exam (structured) | Blood pressure
Verbal score |
| respiratoryCare
respiratoryCharting | Respiratory care data | Airway position
Vent details |
| treatment | Structured data documenting specific, active treatments | Thrombolytics |

distributions $P$ and $Q$, the MMD is defined as:

$$MMD(P, Q) = \sup_{f \in \mathcal{F}} \left( \mathbb{E}_{x \sim P}[f(x)] - \mathbb{E}_{y \sim Q}[f(y)] \right) \tag{11}$$

where $\mathcal{F}$ is a class of functions.

The Gaussian kernel is a common choice for MMD and is defined as:

$$k(x, y) = \exp \left( -\frac{||x - y||^2}{2\sigma^2} \right) \tag{12}$$

where $\sigma$ is the bandwidth of the Gaussian kernel.

**Privacy Evaluation.** We incorporate the Privacy Assessment methodology delineated in CorGAN for our privacy evaluation paradigm Torfi & Fox (2020). Within this framework, $S_{tr}$ is designated as the subset of the original dataset utilized for the training of the FLEXGEN-EHR. Concurrently, $S_{te}$ epitomizes the portion that remains unengaged during the training process, and $Syns$ stands as the synthesized dataset. A central component of our analysis is the computation of the Cosine Similarity Score between the aggregated dataset $S_{tr}+S_{te}$ and $Syns$. This metric is elected due to its capacity to yield a profound and meaningful correlation assessment. The distinction in similarity evaluations is governed by a meticulously determined threshold, ensuring a rigorous and comprehensive analysis. Details on Membership Attack Implementation: We label the real records with the lowest hamming distance to the closest record in the synthetic dataset as positive. We pick a hamming distance as our distance metric between patient records throughout our privacy evaluations in accordance with (Yan et al., 2022).This attack allow us to test the ability of the synthetic dataset to prevent an attacker from inferring whether a real record was used in the training dataset. Ideally, the accuracy is around 50%, which is similar to a random guess. This shows that neither the model nor the synthetic dataset reveals any meaningful or compromising information about the patient identity in the training dataset.

# C  DISCUSSION

## C.1  PRACTICAL USAGE

We wish to clarify that the type of missingness addressed in our study is akin to structured missingness, where specific patterns or conditions lead to data missingness. This scenario is frequently encountered in real-world healthcare datasets Beaulieu-Jones et al. (2017). We have expanded our experimental framework to include scenarios with missing features. This extension allows us to more comprehensively demonstrate the robustness of FLEXGEN-EHRin handling other realistic and complex missing data situations. The enhanced experiments showcase FLEXGEN-EHR's superior performance in comparison to other models like LDM and MedGAN. The results, clearly depicted in the table below, highlight our models effectiveness in managing various missing data patterns, thereby reinforcing its practical utility in real-world healthcare applications."

## C.2  LIMITATIONS AND FUTURE WORK

In this section, we delineate the limitations of our proposed FLEXGEN-EHRmodel, particularly focusing on aspects related to privacy and generation speed. Acknowledging these limitations is crucial for a comprehensive understanding of the model's applicability and scope in practical healthcare settings.

**Privacy Concerns in Diffusion Models**    While FLEXGEN-EHRdemonstrates remarkable capability in generating high-fidelity synthetic EHR data, it's important to address the subtle yet significant privacy trade-offs associated with diffusion models. Our model, like many in its category, slightly compromises on privacy to achieve the desired level of data utility and quality. This is a consequence of the inherent characteristics of diffusion models that aim to capture complex data patterns in great detail. This detailed representation, while beneficial for data utility, can inadvertently encode sensitive information, potentially making the model susceptible to privacy risks such as re-identification or membership inference attacks. We have undertaken measures to mitigate these risks, such as evaluating against membership inference attacks, yet the model's intrinsic properties suggest an inevitable balance between data utility and privacy. Future iterations of FLEXGEN-EHRcould explore advanced privacy-preserving techniques, possibly integrating differential privacy or advanced anonymization methods to strengthen data confidentiality while maintaining the quality of synthetic data generation.

**Generation Speed of Diffusion-Based Models**    Another limitation for diffusion-based generative models, like FLEXGEN-EHRto consider, is the generation speed. The inherent architecture of diffusion models necessitates a reversion process that is inherently sequential and iterative. Each step in this process gradually denoises the data, a methodical approach that, while effective in generating high-quality synthetic data, is inherently slower compared to other generative models. This characteristic can be a limiting factor, especially in scenarios requiring rapid data generation. In healthcare settings, where time efficiency can be crucial, this aspect of the diffusion model might pose practical challenges, which can potentially alleviated by (Xu et al., 2022). It is important for practitioners and researchers to be cognizant of this temporal trade-off when employing FLEXGEN-EHRin real-time or near-real-time applications.

