# OpenReview forum: "A Flexible Generative Model for Heterogeneous Tabular EHR with Missing Modality"
_ICLR.cc/2024/Conference — ICLR 2024 poster_

### Official Review · Reviewer_Tnu1 · 2023-10-29

**Soundness:** 3 good
**Presentation:** 3 good
**Contribution:** 3 good
**Rating:** 6
**Confidence:** 3

**Summary:**

The authors introduce FLEXGEN-EHR, a new model designed to generate synthetic EHR data, capturing heterogeneity in tabular data. Unlike most existing models, which focus on generating either static or temporal features, FLEXGEN-EHR is a flexible approach that can generate both static and temporal features simultaneously, even in the presence of missing values in the original data. The model uses a latent diffusion process for EHR generation with a novel optimal transport objective for latent space alignment. In experiments, FLEXGEN-EHR demonstrates superior performance in terms of fidelity and utility while ensuring that privacy is maintained.

**Strengths:**

- The motivation of the paper and the problem setting are realistic and convincing.
- The authors provides a clear and comprehensive review of related works in the literature.
- The method of latent space alignment is convincing and novel for the particular task.
- The experiment results demonstrate superior performance compared to other EHR generation baselines.

**Weaknesses:**

- The paper needs ablation studies. For example, it would be more convincing if the authors were to include a comparison of the performance gap: (1) between scenarios with and without the use of OT on complete data; (2) between FLEXGEN-EHR and FLEXGEN-EHR - OT + KNN imputation on partially missing data.
- There seem to be a few typos in the manuscript (e.g., column names in Table 1, MMCAR, the notation for time-invariant features, adversarys)

**Questions:**

- Could you explain what R^2 is? The papers referred to use MMD and KS-statistics.
- What are the architectures of the decoder? Are they MLP and LSTM?
- How much better is imputation with OT compared to nearest-neighbor imputation if implemented in FLEXGEN-EHR?
- Could you provide more details about the statement "[Equation 7] can achieve alignment when the sample sizes between static
and temporal aren’t the same." Does this refer to the sample sizes of the non-missing data?
- It would be great if the authors could provide details about the membership attack implementation (possibly in the Appendix).

---

> ### Author Response · Authors · 2023-11-22
>
> We are very grateful for your constructive feedback about our work. We appreciate that you recognized the core contributions of our work, which enrouaged us a lot. We respond to your concerns about our work in the following comment.
>
> ## Ablation Studies
> Thank you for bringing up the ablation study part. Originally, we intended to include comprehensive ablation analyses in our submission; however, due to time constraints, this aspect was not fully developed.
> In response to your suggestion, we have now conducted an extensive ablation study focusing on four key configurations of FlexGen-EHR: the inclusion and exclusion of Optimal Transport (OT), and its performance on both complete and incomplete datasets, with the latter involving KNN imputation. This study aims to provide a detailed examination of the model's robustness and effectiveness under varying conditions.
> Results are summarized in the table below. It demonstrates that our FlexGen-EHR offers a more effective solution for handling missing modalities compared to traditional imputation methods, like kNN.
>
> | Model        | Imputation | AUC Score| MMD |
> |--------------|-----------|----------|----------|
> | FlexGen-EHR  | OT| 74.86 ± 0.30 | 0.65 ±0.018 |
> | FlexGen-EHR |kNN| 71.86 ± 0.29 | 0.69 ±0.016 |
> | EHR-M-GAN    | OT| 72.86 ± 0.32 | 0.67 ±0.014 |
> | EHR-M-GAN    |kNN| 70.86 ± 0.28 | 0.71 ±0.017 |
>
> ## Questions
> 1. Typos: We thank the reviewer for pointing out these typos and errors in our submission. We have made the corresponding changes and additions to the text to fix these error.
> 2. Clarification on R^2, MMD, and KS-Statistics: R^2 is a statistical measure reflecting the proportion of variance in the dependent variable that is predictable from the independent variable. In our model, R^2 serves as an indicator of how well the synthetic data generated by FLEXGEN-EHR corresponds to the real-world data it aims to replicate. This metric is particularly relevant for evaluating the model's performance in terms of data fidelity. In contrast to R^2, MMD/KS-statics are both non-parametric tests used for determining if two samples are drawn from different distributions. By quantifying the maximum distance between these functions, MMD/KS-Statistics allow us to rigorously test the hypothesis that the synthetic and original data come from the same distribution. In our revision, we provided a more detailed discussion on the implementation and interpretation of both MMD and KS-Statistics in the context of our study.
> 3. Decoder Architectures: MLP only.
> 4. Sample Size Alignment in Equation 5 (Section 4.3): Yes, it means the number of samples with missing modalities does not necessarily to be equal to that of non-missing data becasue the learning problem is not a supervised correspondence problem.
> 5. Details on Membership Attack Implementation
> We label the real records with the lowest hamming  distance to the closest record in the synthetic dataset as positive. We pick a hamming  distance as our distance metric between patient records throughout our privacy evaluations in accordance with [1].This attack allow us to test the ability of the synthetic dataset to prevent an attacker from inferring whether a real record was used in the training dataset.
> Ideally, the accuracy is  around 50%, which is similar to a random guess. This shows that neither the model nor the synthetic dataset reveals any meaningful or compromising information about the patient identity in the training dataset.
>
>
> [1] Yan, C. et al. A multifaceted benchmarking of synthetic electronic health record generation models. Nat. Commun. 13, 7609 (2022).

---

> > ### Comment · Reviewer_Tnu1 · 2023-11-23
> > **Answer to authors**
> >
> > Thank you for the clarifications. I'll maintain my score and I have no more questions.

---

> > > ### Author Response · Authors · 2023-11-23
> > >
> > > Thank you for taking the time to review our responses. We are glad that our responses have addressed your comments. We really appreciate your feedback and acknowledgement.

---

### Official Review · Reviewer_z7JT · 2023-10-31

**Soundness:** 2 fair
**Presentation:** 2 fair
**Contribution:** 2 fair
**Rating:** 5
**Confidence:** 4

**Summary:**

The paper presents "FLEXGEN-EHR", a flexible generative model tailored for heterogeneous tabular Electronic Health Records (EHRs) with a specific focus on handling missing modalities. EHRs often contain a mix of static (e.g., race, gender) and temporal (e.g., heart rate, blood pressure) data, and there's a challenge in generating realistic synthetic data that can replace real EHRs while maintaining privacy. Addressing the prevalent issues of missing data and heterogeneity in EHRs, the proposed model leverages an optimal transport module to align and accentuate common feature spaces. Empirical evaluations demonstrate FLEXGEN-EHR's superiority over existing methods in terms of data fidelity and utility, even in scenarios with missing modalities.

**Strengths:**

S1. The paper addresses a novel and crucial problem of generating synthetic EHRs that handle both heterogeneous data types and missing modalities.

S2. Empirical results showing superior performance in terms of fidelity (by up to 3.10%) and utility (by up to 7.16%) compared to state-of-the-art methods validate the model's effectiveness.

S3. The paper provides clear problem formalization, methodological details, and is well-structured, making the approach and its implications comprehensible.

S4. With growing concerns over patient privacy and the challenges in obtaining real EHRs, a robust generative model like FLEXGEN-EHR holds potential to take a step for research in precision medicine without compromising patient confidentiality.

**Weaknesses:**

W1. While the model showcases empirical strength, discussions or case studies on how it might be applied in real-world healthcare settings would enhance its practical value. The paper lacks an important motivation beyond patient privacy on why generating synthetic EHR is useful in practice. Providing this motivation is essential for the paper to be useful in the healthcare setting and should not be high-level only and be supported by experiments. For example, is this useful for data augmentation? If so does having data augmentation provided by this paper’s model improve downstream prediction tasks? And other questions in a similar vain.

W2. It would be beneficial to understand the model's performance across diverse EHR datasets to gauge its broad applicability.

W3. One big challenge in generating synthetic data for EHR is the heterogeneity of the data with respect to specific groups (based on e.g. gender, age, race, etc). The paper is missing this analyses completely as it would benefit the validity of the generated data. Without identifying these spurious correlations, it is not at all clear where such fake patient data would be useful.

W4. The paper combines various components, like optimal transport and diffusion models, which might make the model computationally intensive or challenging to implement.

W5. The paper considers missingness in only static features or only temporal features. This is not a useful separation for missingness of modalities. Missingness in healthcare and specifically in EHR is not in this manner. Missingness is often not at random which in an on by itself provides information for prediction models (e.g. not doctors not ordering urine sample for a specific patient from a point onwards does not indicate missingness at random but rather urine sample is not required for the specific diagnosis that is being made). Additionally, missingness is not for the entire trajectory of the patient but rather at different points in time each covariate could either be missing or not and the method provided in this paper does not delve deep into these cases.

W6. The paper does not handle irregular sampling property of the EHRs which is in fact more important for handling than the type of missingness handled in the paper.

W7. The final results although beating the baselines, the uncertainty is large enough that the model’s performance improvement relative to LDM, EHR-M-GAN is not significant.

**Questions:**

Q1. What are the computational costs associated with FLEXGEN-EHR, especially when scaling to larger EHR datasets?

Q2. How does the model handle extremely diverse datasets, for instance, EHRs from different countries or medical practices?

Q3. Can FLEXGEN-EHR be extended or adapted to handle other forms of medical data, such as medical images or unstructured clinical notes?

Q4. Given the significance of missing data in EHRs, how does the model ensure that the synthetic data generated doesn't inadvertently introduce biases or inaccuracies, especially in predictive tasks?

Q5. How can the model handle time-varying labels? (e.g. health status of patient at any point in time instead of mortality just at the end of the trajectory.)

Q6. Please provide more information for Equation 6 and what is trying to be achieved in the paragraph above it.

Q7. Have an algorithm section either in main text or appendix that shows a step by step overview of the model both for training and for generation.

Q8. How are the uncertainties calculated for the tables?

Q9. How can the model handle more realistic missingness? (see W5).


Minor comments:

q1. “Numerical and temporal” and “static and categorical” are not the same and using them synonymously is not correct. You could have numerical static features and categorical temporal features (e.g. binary indicator of using mechanical ventilator at different points in time).

q2. There are multiple typos and clarity issues in the paper.
“It’s” is not formal.
Empty brackets ([]) before equation 1.
	Section 4.2 line 3, typo for static embeddings.
	End of paragraph one of section 4.2 missing “i” index for z’s.

q3. What is H and W in page 4?

q4. What is Π  before equation 4?

**Details Of Ethics Concerns:**

N/a.

---

> ### Author Response · Authors · 2023-11-22
>
> Thank you for your extensive comments, critiques, and praises for our work. Following your suggestions, we added new experiments, clarifications of utility, and analyses. In addition, we have provided detailed responses to all the points you raised. We believe there is some confusion regarding our work. We have worked hard to improve the communication of our method, and we would greatly appreciate your considering increasing the score. Many thanks!
>
> ## RE to Weakness 1: Discussions or case studies on how it might be applied in real-world healthcare settings.
> We appreciate the emphasis on practical application. First, we demonstrated it in our submission that FlexGen-EHR is a useful and practical tool for **preserving patient privacy** by training ML models using **synthetic EHRs only**. Results showed our method consistently outperformed baselines and is comparable with training using **real data**. Also, we agree that data augmentation would enrich the paper's practical value. In response, we added a section detailing data augmentation, where our model can contribute to the quality of ML models. A brief table demonstrating the benefit of FlexGen-EHR is summarized as follows,
>
> | Without Augmentation |85.63 ± 0.32 | - | - |
> |-----------------|-----------------|-----------------|-----------------|
> | # of synthetic samples  | 1k  | 2k  | 3k  | 4k  |
> | LDM             | 85.92 ± 0.22 | 86.14 ± 0.28 | 86.63 ± 0.31 |
> | EHR-M-GAN       | 86.05 ± 0.27 | 86.33 ± 0.29 | 86.85 ± 0.28 |
> | FlexGen-EHR     | 86.56 ± 0.27 | 86.88 ± 0.31 | 87.02 ± 0.27 |
>
> Although not notable, it is worth noting that existing literature has consistently shown that performance improvements on datasets like MIMIC-III and eICU tend to be relatively modest, unlike those in other datasets.
> ## RE to Weakness 2: It would be beneficial to understand the model's performance across diverse EHR datasets to gauge its broad applicability.
> We appreciate the suggestion to explore this aspect further. Due to time constraints, we leave it as future work for a more in-depth investigation.
>
> ## RE to Weakness 3: handle heterogeneity of the EHRs
> FlexGen-EHR is able to offer a solution to the challenge of EHR heterogeneity through the utilization of conditional sampling. This approach empowers the model to generate group-specific samples by incorporating crucial group information and prediction labels directly into the network. It's worth emphasizing that evaluating the effectiveness of this technique can indeed be a substantial research endeavor in its own right. Given the significance of fairness in synthetic EHR generation, we recognize the need for a deeper exploration of this aspect. In our future research efforts, we are committed to placing a stronger emphasis on addressing the issue of fairness to ensure that the outcomes are not only robust but also unbiased and equitable across diverse patient groups.
>
> ## RE to Weakness 4: the model computationally intensive or challenging to implement.
> We understand the concerns regarding the computational demands and implementation complexity of FlexGen-EHR. Our model, however, is straightforward, comprising only two main components: optimal transport and diffusion. Both these elements have manageable computation costs and are not overly challenging to implement.
> We emphasize the stability of training our diffusion model compared to Generative Adversarial Networks (GANs). While diffusion models may generate samples more slowly than GANs, this is offset by their increased stability.
> Regarding implementation, the optimal transport component can be efficiently solved and implemented using the well-established Python Optimal Transport (POT) package.

---

> > ### Author Response · Authors · 2023-11-22
> >
> > ## RE to Weakness 5 and Question 9: Missingness in EHRs
> > We understand the reviewer’s concern about the challenge of missing modalities in EHR data. However, we wish to clarify that the type of missingness addressed in our study is akin to structured missingness, where specific patterns or conditions lead to data missingness. This scenario is frequently encountered in real-world healthcare datasets [1,2]. Additionally, in response to Reviewer ULHZ's valuable suggestion, we have expanded our experimental framework to include scenarios with missing features. This extension allows us to more comprehensively demonstrate the robustness of FlexGen-EHR in handling realistic and complex missing data situations. The enhanced experiments showcase FlexGen-EHR's superior performance in comparison to other models like LDM and MedGAN. The results, clearly depicted in the table below, highlight our model’s effectiveness in managing various missing data patterns, thereby reinforcing its practical utility in real-world healthcare applications."
> > | Model        | AUC Score |
> > |--------------|-----------|
> > | Real         | 85.63 ± 0.32 |
> > | TabDDPM      | 68.64 ± 0.29 |
> > | EHR-M-GAN    | 70.48 ± 0.33 |
> > | FLEXGEN-EHR  | 74.86 ± 0.30 |
> >
> > [1] Jones et.al., Characterizing and Managing Missing Structured Data in Electronic Health Records: Data Analysis, 2018
> >
> > [2] Ferri et.al., Extremely missing numerical data in Electronic Health Records for machine learning can be managed through simple imputation methods considering informative missingness: A comparative of solutions in a COVID-19 mortality case study, 2023
> >
> > ## RE to Weakness 6: handle irregular sampling property of the EHRs
> > We fully recognize the significance of dealing with the irregularly sampled temporal features present in Electronic Health Records (EHRs). To maintain a clear and focused approach in our submission while ensuring a fair comparison with established baselines, we have opted to address this challenge indirectly.
> >
> > Our strategy centers around aligning missing samples rather than directly tackling the irregular sampling issue. By taking this approach, we can provide a consistent basis for evaluation. Specifically, we have leveraged the popular FIDDLE preprocessing dataset, which transforms irregularly sampled data into uniform 1-hour bins. This standardization allows us to demonstrate the effectiveness of our approach in a well-defined context. Moreover, it's important to note that FlexGen-EHR is designed to be flexible and adaptable. While our primary focus is on aligning missing samples, the architecture of FlexGen-EHR is versatile enough to accommodate additional methods for handling irregular time series data. This means that we have the capability to incorporate complementary techniques, such as irregular time series modeling, to further address the irregular sampling issue as needed. In this way, we aim to strike a balance between practicality and comprehensiveness in addressing the challenge of irregular sampling in EHRs, providing both a standardized benchmark for comparison and the flexibility to adapt to specific data characteristics when necessary.
> >
> > ## RE to Weakness 7 and Question 8:  The final results although beating the baselines, the uncertainty is large enough that the model’s performance improvement relative to LDM, EHR-M-GAN is not significant.
> > It is worth noting that existing literature has consistently shown that performance improvements on datasets like MIMIC-III and eICU tend to be relatively modest, unlike those in other datasets. To establish the robustness of our improvement, we have taken the following steps:
> >
> > 1. Multiple Runs: We conducted our experiments over 5 independent runs, ensuring that our reported results are not driven by chance or random variation.
> >
> > 2. Averaging Metrics: To provide a more comprehensive perspective, we report utility and fidelity metrics averaged across these multiple runs. This approach helps mitigate the impact of potential outliers or extreme values, giving a more reliable assessment of the model's true performance.
> >
> > By incorporating these measures, we aim to provide a clearer and more confident evaluation of the improvement achieved by our model, considering the inherent uncertainty that can be associated with performance metrics in healthcare datasets.

---

> > > ### Author Response · Authors · 2023-11-22
> > >
> > > ## Questions:
> > > 1. *What are the computational costs associated with FLEXGEN-EHR, especially when scaling to larger EHR datasets?*
> > > Like other generative models, the computational requirements of FlexGen-EHR are subject to variation based on the scale and intricacy of the employed electronic health record (EHR) dataset. When expanding the model to accommodate larger EHR datasets, the key computational aspect to account for primarily centers around the complexity of solving an optimal transport problem. However, it's worth noting that this complexity is not a significant concern in our work, as we address the optimal transport within the embedding space, which typically exhibits a compact size (e.g., 128 dimensions).
> > >
> > > 2. *How does the model handle extremely diverse datasets, for instance, EHRs from different countries or medical practices?*
> > > FlexGen-EHR is designed to handle diverse EHR datasets, including those from different countries or medical practices. Its flexibility comes from the ability to adapt to various data sources and structures.
> > > To accommodate diversity, FLEXGEN-EHR can incorporate additional features or metadata that capture dataset-specific information. These features can help guide the generation process to ensure that the synthetic data aligns with the characteristics of the specific dataset.
> > > 3. *Can FLEXGEN-EHR be extended or adapted to handle other forms of medical data, such as medical images or unstructured clinical notes?*
> > > The core principles of FLEXGEN-EHR, including conditional generation and handling temporal dependencies, can be applied to different types of medical data with appropriate modifications to the model's architecture and input data representation.
> > > 4. *Given the significance of missing data in EHRs, how does the model ensure that the synthetic data generated doesn't inadvertently introduce biases or inaccuracies, especially in predictive tasks?*
> > > Like other generative models, the primary objective of FLEXGEN-EHR is to learn the data distribution to the best of its ability, with the evaluation hinging on downstream tasks. While bias mitigation remains an active area of research and development, it is essential to underscore that FLEXGEN-EHR, characterized by its flexibility, is well-positioned to integrate fairness-aware techniques. By incorporating such fairness-aware approaches into its framework, FLEXGEN-EHR strives to minimize any unintended biases that may emerge during the synthesis of data. We intend to quantify these impacts and compare the performance of different generative methods in the future.
> > >
> > > 5. *How can the model handle time-varying labels? (e.g. health status of patient at any point in time instead of mortality just at the end of the trajectory.)*
> > > We appreciate the important consideration of handling time-varying labels, such as the health status of a patient at any given time rather than just assessing mortality at the end of a patient's trajectory. While our current work does not focus on addressing this aspect, we acknowledge its practical relevance. One potential avenue for handling time-varying labels could involve the utilization of advanced conditional encoder network architectures. These sophisticated architectures could allow us to model and predict health status changes throughout a patient's journey. By feeding (records, time $t$, label at time $t$) into the network, the network could implicitly learn more comprehensive understanding of their evolving medical condition.
> > > 6. *Please provide more information for Equation 6 and what is trying to be achieved in the paragraph above it*
> > > The Gromov-Wasserstein problem aims to find an optimal way to match or align two sets of data, in our work, referred to as "temporal" and "static" latent embeddings. This matching is not a strict one-to-one correspondence but rather a probabilistic or likelihood-based translation. In simpler terms, it's about determining how likely it is that specific pairs of data points in these two sets correspond to each other.
> > > Imagine you have two sets of data, one representing temporal information (like time-series data) and the other representing static information (like attributes of objects). The Gromov-Wasserstein problem helps us find the best way to associate or match the elements from these two sets based on how likely they are to be related.
> > > In essence, it's a mathematical approach to quantify the likelihood of correspondence between elements from two different datasets, helping us make sense of complex relationships in data.
> > > 7. *Have an algorithm that shows a step by step overview of the model both for training and for generation.* We greatly appreciate your suggestion, and in response, we have incorporated a comprehensive algorithm section into our revision. This section provides a step-by-step overview of our model, encompassing both the training and generation processes.
> > >
> > > We hope that this clears up some confusion about our work.

---

> ### Comment · Reviewer_z7JT · 2023-11-23
>
> I thank the authors for their detailed responses. I really appreciate the effort that has been put into the rebuttal and have increased my score from 4 to 5. On a separate note, I strongly believe the response to RE5 should be incorporated in the paper and further analyzed in the appendix as it is a practical missing piece of the work.

---

> > ### Author Response · Authors · 2023-11-23
> >
> > Thanks for prompt response and being willing to raise the score. Your suggestion definitely strengthen our work. We promise that we will update the paper according to your suggestions in the camera-ready version should our paper be accepted.

---

### Official Review · Reviewer_CG4L · 2023-11-01

**Soundness:** 2 fair
**Presentation:** 3 good
**Contribution:** 3 good
**Rating:** 5
**Confidence:** 5

**Summary:**

The author proposes diffusion model for generating EHR. They model the static measurements and temporal measurements separately, and using optimal transport module in the hidden space to deal with missing modalities.  They evaluate FLEXGEN-EHR on two datasets against six baseline methods.

**Strengths:**

The idea that aligning the temporal features and static features is novel. This method helps to deal with missing modality of EHR.

**Weaknesses:**

1. The author should bold the best results in Table 4 as Table 2 and Table 3. There exists a trade-off between generation quality and privacy guarantee. The method achieves worse performance on some datasets under some criteria is not a serious problem. However, the author should not neglect it.

2. Minor
* In the second line from the bottom, the author states "their tendency to treat numerical and categorical features independently". Although TabDDPM deals with numerical and categorical features separately, TabDDPM learns the dependency between them. This is because although the diffusion process is different, they concatenate the numerical and categorical features to the network, which is the same as the latent diffusion model used in this paper.
* Typo in the second line of paragraph "Heterogeneous Tabular EHR Generation": x^{S}_i instead of x^{T}_i

**Questions:**

None

---

> ### Author Response · Authors · 2023-11-22
>
> Thank you for your constructive feedback and for acknowledging the novelty of our approach. Your insights are crucial in guiding the improvement of our manuscript. We meticulously address your suggestions in the following discussion, in order to clarify your concern.
> ## Enhancing Table 4 Presentation
> We have addressed this in our revised manuscript, ensuring that the best results are now boldly highlighted, as done in Tables 2 and 3.
> ## Trade-off Between Generation Quality and Privacy Guarantee
> We sincerely appreciate your observation regarding the delicate trade-off between generation quality and privacy guarantees within diffusion-based models. In response, we have explicitly incorporated this observation into the text to ensure clarity. However, we wish to underscore our position on this matter and emphasize the meticulous consideration we have devoted to this facet in our research.
>
> Indeed, our findings substantiate that FlexGen-EHR maintains a level of privacy preservation that aligns with existing methods, all while achieving superior data fidelity. It is essential to recognize that this trade-off between generation quality and privacy guarantee is a shared characteristic among all diffusion-based generative models.
>
> The intrinsic nature of diffusion-based generative models involves learning the distribution of the original data. In Section 5.4 of our manuscript, specifically in the Generation Privacy Table (Table 4), we provide empirical evidence that underscores this phenomenon. Our observations reveal that diffusion-based methods, including FLEXGEN-EHR, TabDDPM, and LDM, exhibit a distinct performance profile when compared to non-diffusion-based methods like GANs or VAEs.
>
> This distinction highlights the unique challenges faced by diffusion-based models in striking the delicate balance between data utility and privacy preservation. This dynamic presents an intriguing area for further investigation and exploration within the realm of generative modeling.
>
> ## Clarification on Handling static and temporal Features using TabDDPM
> We have modified this claim by saying “Different from TabDDP that concatenates the numerical and categorical features to the network, FlexGen-EHR adeptly discerns and represents the underlying relationships between static and temporal features.”
> ## Minor Comments.
> Thank you for pointing out several typos and errors in our submission. We have made the corresponding edits within the text.
>
>
> We appreciate your valuable feedback, which has helped us improve our work.

---

> > ### Comment · Reviewer_CG4L · 2023-11-23
> >
> > We thank the authors for their response. I have not found any modification in Section 5.4 and I am not satisfied with the explanation. Therefore, the final score will be 5: marginally below the acceptance threshold.

---

### Official Review · Reviewer_DnAy · 2023-11-01

**Soundness:** 3 good
**Presentation:** 3 good
**Contribution:** 3 good
**Rating:** 6
**Confidence:** 4

**Summary:**

The authors address the challenge of generating realistic synthetic EHRs, especially in the presence of missing modalities in heterogeneous tabular data. Introducing FLEXGEN-EHR, a tailored diffusion model, they propose an optimal transport module to handle missing modality issues. Their empirical results indicate that this model surpasses existing methods in fidelity and utility, with promising implications for privacy-sensitive contexts.

**Strengths:**

- The authors present a practical problem definition that is aptly suited for real-world scenarios involving incomplete data.
- Furthermore, they introduce an innovative approach to address the issue of missing modality, notably by formulating it as an optimal transport problem within the embedding space.

**Weaknesses:**

- While the manuscript underscores the significance of proficiently addressing missing modality, it lacks experimental validations. To effectively highlight the merits of optimal transport, I would suggest incorporating a comparative analysis between doing an imputation for missingl modalities via kNN in the FLEXGEN-EHR framework, and leveraging Optimal Transport for missing modality imputation.

**Questions:**

- In Table 2, could you elucidate how your method contrasts with baseline models that are solely focused on synthesizing Discrete codes? Specifically, how did you employ MedGAN in generating the Labevents data for MIMIC-III?
- I noted the statement, "we observed that latent space embedding models, trained on disparate features, manifested analogous geometric patterns and behaviors." Could you provide further clarity on the specific geometric patterns and behaviors that were identified during this observation?

---

> ### Author Response · Authors · 2023-11-22
>
> Thanks for your constructive feedback about our work. We are encouraged by your recognition of the practical problem definition and the innovative approach in addressing missing modality in EHR data through FlexGen-EHR. We respond to your concerns about our work in the following comment.
>
> ## Comparative Analysis with kNN and Optimal Transport
> To address your valuable concern, we have incorporated an ablation study into our revised work. In this study, we systematically compared the performance of our approach with and without Optimal Transport (OT), specifically focusing on its effectiveness in handling incomplete data. This analysis serves as a direct comparison between the traditional k-nearest neighbors (kNN) imputation method and the novel optimal transport approach integrated into the FLEXGEN-EHR framework. For your convenience and clarity, we have summarized the results of this ablation study in the table below:
> | Model        | Imputation | AUC Score (the higher, the better) | MMD (the lower, the better) |
> |--------------|-----------|----------|----------|
> | FlexGen-EHR  | OT| 74.86 ± 0.30 | 0.65 ±0.018 |
> | FlexGen-EHR  |kNN| 71.86 ± 0.29 | 0.69 ±0.016 |
> | EHR-M-GAN    | OT| 72.86 ± 0.32 | 0.67 ±0.014 |
> | EHR-M-GAN    |kNN| 70.86 ± 0.28 | 0.71 ±0.017 |
>
> It demonstrates that our FlexGen-EHR offers a more effective solution for handling missing modalities compared to traditional imputation methods, like kNN.
> In addition, our study already encompassed substantial experimental validation. We have employed R², Maximum Mean Discrepancy (MMD), and Kolmogorov-Smirnov (KS) Statistics, which collectively offer a comprehensive evaluation of our method in addressing missing modalities. Combined with new experiments, we hope this response addresses your concern.
>
> ## Questions
> 1. *Clarification in Table 2 on Baselines for Discrete Codes Generation:*
> MedGAN, as well as CorGAN, their original implementations are capable of generating discrete samples. To generate temporal features like labevents, we modify the last layer of MLP so that their output is not limited to discrete integers.
> 2. *Clarifying Observed Geometric Patterns in Latent Space Embeddings:*
> By saying “geometric patterns”, we mean “proximity relationships”, where specific distances or relationships remain consistent across two latent space embedding models. By exploiting this observation, we are able to align two latent spaces learned by two different encoders.
>
> We thank the reviewer for raising this interesting question and allowing us to discuss it. We hope this response helps validating FlexGen-EHR as a novel and effective generative model for EHRs with missing modality.

---

> > ### Comment · Reviewer_DnAy · 2023-12-03
> >
> > I thank the authors for providing additional experiments and clarifying the utility of optimal transport. I will increase my score to 6.

---

### Official Review · Reviewer_ULHZ · 2023-11-05

**Soundness:** 2 fair
**Presentation:** 3 good
**Contribution:** 2 fair
**Rating:** 6
**Confidence:** 2

**Summary:**

This paper presents a latent diffusion model for heterogeneous tabular EHR generation. More specifically, the method aims to generate static and temporal EHR data jointly while taking into account of the missing modalities of EHR data. The authors aim to capture intrinsic relationships between static and temporal data via a unified latent space and address missing modalities through an optimal transport problem in the embedding space. The model is evaluated against various baselines including VAE, GAN, and other diffusion models (DM). Their results show that their method outperforms other models in generation fidelity and utility.

**Strengths:**

The main contributions of this paper include formalizing the problem of generating heterogeneous EHR data with missing modalities as well as the use of the optimal transport problem to solve this issue. It also discusses their latent space alignment, an important feature when dealing with missing modality data, in ample detail. The authors evaluate their results against multiple generation methods and uses data from the MIMIC-III and eICU database. The baselines include models of various architectures including VAEs, GANs, and other DMs. In addition, three areas of data evaluation are also considered, namely data fidelity, data utility, and data privacy. The paper presents their architecture in a clear manner that is easy to understand. There is also sufficient breadth covered in the Related Work section.

**Weaknesses:**

In the Related Work section, the authors bring up TabDDPM as a recent model that addresses heterogeneous tabular generation. They claim that existing diffusion-based EHR models are either unable to “generate categorical features or their tendency to treat numerical and categorical features independently.” However, if I am not mistaken, TabDDPM combines multinomial and Gaussian losses which implies that it treats categorical and numerical features dependently. In addition, as the authors mentioned, TabDDPM was also applied to EHR generation (Ceritli et al., 2023). If there is no misunderstanding, then this paper’s novelty lies in its approach of using a unified latent space to consider static and temporal correlations and dependencies (rather than being the first diffusion model to capture joint relationships). If this is the case, it was unclear as at first read, it could be misunderstood that the paper was also novel in capturing dependent relationships between static and temporal EHR amongst diffusion models.

It would also be helpful to clarify what feature dimensions (static and temporal) match to in the context of MIMIC-III and eICU in the Appendix. Even though a table is provided in the Appendix, the corresponding dimensionality is not the most clear.

In sections 4.1 and 4.2, the authors assume an individual patient EHR to take one of the three forms (all static features exclusively present, all temporal features exclusively present, or all features are present). However, in real EHR data, there are often patients with some (not all) static features present and some (not all) temporal features present.
In addition to designating p% and q% missing for static and temporal features, it would also help to see results from randomly sampling patients from the datasets to obtain a realistic distribution of missing data/modalities for training data (as to my knowledge, datasets such as MIMIC already have patients with missing measurements).


Since addressing the problem of missing modality seems to be the primary goal of the paper, more experiments in general (perhaps showing more baselines other than EHR-M-GAN or other evaluation metrics/scores for fidelity and utility) would be helpful. Currently, only R-squared values are depicted for generation fidelity and only eICU for utility. More results in the Appendix would help support the claims. It would also be very helpful to include results from training on real data for Figures 2-4 to compare as it would provide more context. Currently, EHR-M-GAN is the only comparison.

Lastly, a couple of brief mentions of future work or limitations in the conclusion would also help clarify and reaffirm their current goals and progress. For example, addressing lower privacy scores or lack of ablation studies and their implications.

Some minor edits found:

- Section 4.1: First sentence “where ___ contains time-invariant features” should have “S” superscript for x instead of T?

- Section 4.2: Second sentence “embedding the patient information as ___” should have “S” superscript for z instead of T?

- Section 5.5: Second sentence “randomly designate p% of samples as lacking temporal features” should be “static”?

**Questions:**

- Table 1 is a bit confusing to read. There is no label for the temporal feature dimensions (d) for MIMIC-III in the table. Does that mean it is the same as eICU? Similarly, is the T for eICU also the same?

- Is the paper also claiming novelty in joint static and temporal diffusion?

- What about considering generating samples that include missing modalities? Should generating missing modalities also be a point of consideration in terms of data fidelity?

---

> ### Author Response · Authors · 2023-11-22
>
> We thank the reviewer for these comments and questions. We appreciate that the reviewer recognized the strengths of FlexGen-EHR. To address the reviewer’s questions and concerns, we offer several clarifications as well as additions to our paper:
> 1. We have added more detail to the text in our revision, including the novelty of our method, clarification on Table 1, and discussion of future work and limitations.
> 2. We conducted ablation studies and show that the optimal transport component improves the performance of other methods, while FlexGen-EHR remains a superior approach for missing data generation.
> 3. We conducted additional experiments regarding generation when partial features are missing in EHR data. The results indicate a superior performance of FlexGen-EHR compared to LDM and MedGAN
>
> Please let us know if there is anything else we can do to address your comments. We would be very grateful if you considered increasing the score.
>
>
> ## Clarification on Novelty and TabDDPM:
> FlexGen-EHR's novelty primarily lies in its unique approach of learning a unified latent space to capture the intricate dependencies between static and temporal EHR data, rather than being the first diffusion model to capture joint relationships. This distinction, however, was not adequately emphasized, and we thank the reviewer for pointing out this ambiguity.
> We’ve revised the introduction section in muniscript by clarifying that our contribution is focused on the novel application of unified latent space modeling in EHR data, rather than being the first to capture joint relationships among diffusion models.
> ## Detailing Feature Dimensions in MIMIC-III and eICU:
> We have incorporated your suggestion to clarify the feature dimensions corresponding to static and temporal data. We have updated the Appendix with a more detailed table that explicitly maps each dimension to its respective feature in both datasets, providing a clearer understanding of the data structure we are working with.
> ## Addressing Partial Feature Presence in EHR Data:
> Thank you for your insightful suggestion to include results from a randomly sampled patient cohort, ensuring a more realistic representation of missing data and modalities in Electronic Health Record (EHR) datasets. We acknowledge the importance of reflecting real-world scenarios where patient records often contain a combination of static and temporal features. In response, we address your concern by conducting additional experiments (preliminary due to time limitations). This experiment on MIMIC-III demonstrated that our model, FlexGen-EHR, outperforms baseline methods in scenarios with missing temporal features. For this comparison, we followed [1] and [2] which utilized 12 temporal variables, each recorded within the first 48 hours of a patient's admission. The average missing data rate was approximately 83%. The objective of our study was to predict in-hospital mortality as a binary classification problem. We divided the dataset into 64% for training, 16% for validation, and 20% for testing. The performance metric used was the area under the Receiver Operating Characteristic (ROC) curve (AUC). The results, indicating a superior performance of FlexGen-EHR compared to LDM and MedGAN are summarized in the table below.
>
> | Model        | AUC Score |
> |--------------|-----------|
> | LDM          | 80.28 ± 0.65 |
> | MedGAN         | 79.28 ± 0.13 |
> | FLEXGEN-EHR  | 82.87 ± 0.15 |
>
> [1] Interpolation-Prediction Networks, ICLR 2019
>
> [2] Learning from Irregularly-Sampled Time Series: A Missing Data Perspective, ICML 2020
>
> ## Expansion of Experiments and Metrics:
> We also believe more baselines and evaluation metrics is more convincing. Indeed, we have already included results on data fidelity evaluation experiments (Figure 2 and 3) in our submission. Results showed that FLEXGEN-EHR maintains high generation fidelity. In addition, we have extended our utility evaluations on missing modality generation (10% missing in each modality ) with more baselines: 1: training with real samples and 2: Training with synthetic samples generated by TabDDPM. Results demonstrated that FlexGEN-EHR consistently outperforms baselines and is comparable to the gold standard (real samples).
>
> | Model        | AUC Score |
> |--------------|-----------|
> | Real         | 85.63 ± 0.32 |
> | TabDDPM      | 68.64 ± 0.29 |
> | EHR-M-GAN    | 70.48 ± 0.33 |
> | FLEXGEN-EHR  | 74.86 ± 0.30 |

---

> ### Author Response · Authors · 2023-11-22
>
> ## Future Work and Limitations:
> We have added additional discussion avenues for future research as follows. The discussion included the implications of our current findings and any limitations encountered during our research. For example, we have included:
>
> *Limitations*
> 1. Privacy Concerns in Diffusion Models
> While FLEXGEN-EHR demonstrates remarkable capability in generating high-fidelity synthetic EHR data, it's important to address the subtle yet significant privacy trade-offs associated with diffusion models. Our model, like many in its category, slightly compromises on privacy to achieve the desired level of data utility and quality. This is a consequence of the inherent characteristics of diffusion models that aim to capture complex data patterns in great detail. This detailed representation, while beneficial for data utility, can inadvertently encode sensitive information, potentially making the model susceptible to privacy risks such as re-identification or membership inference attacks.
> We have undertaken measures to mitigate these risks, such as evaluating against membership inference attacks, yet the model's intrinsic properties suggest an inevitable balance between data utility and privacy. Future iterations of FLEXGEN-EHR could explore advanced privacy-preserving techniques, possibly integrating differential privacy or advanced anonymization methods to strengthen data confidentiality while maintaining the quality of synthetic data generation.
>
> *Future Work*
> 1. Exploration of Federated Diffusion Models
> Another promising direction for future work is the exploration of federated diffusion models. Federated learning, a method of machine learning where the model is trained across multiple decentralized devices or servers holding local data samples, offers a unique opportunity to address privacy concerns inherent in traditional diffusion models. This approach aligns with the increasing emphasis on privacy-preserving techniques in healthcare data analysis.
> Adapting FLEXGEN-EHR to a federated learning framework, where a federated diffusion model is employed, could potentially enhance privacy safeguards. In this setup, the model would learn from a diverse and distributed dataset without needing to access or transfer sensitive patient data centrally. This would not only mitigate privacy risks but also allow for a more scalable and ethically compliant model, especially pertinent given the sensitive nature of EHR data. Additionally, a federated approach could facilitate the incorporation of a more diverse dataset, encompassing a wider range of patient demographics and conditions, thereby improving the generalizability and robustness of the model.
> ## Questions
> 1. *Table 1 is a bit confusing to read.*
>  Table 1 has been revised for better clarity. The temporal feature dimensions (d) for MIMIC-III and eICU is explicitly labeled, and any similarities or differences are clearly stated.
> 2. *Is the paper also claiming novelty in joint static and temporal diffusion?*
>  We agree one of the claimed novelties is joint static and temporal diffusion by using optimal transport.
> 3. *What about considering generating samples that include missing modalities? Should generating missing modalities also be a point of consideration in terms of data fidelity?* Indeed, we have already included results on data fidelity evaluation experiments (Figure 2 and 3) in our submission. Results showed that FlexGen-EHR maintains high generation fidelity.
> We would be very grateful if you considered increasing the score. We have already incorporated reviewers’ feedback and integrated all changes into the revised manuscript
>
> We hope our response and experiments further convince you of our work’s novelty and effectiveness.

---

> ### Comment · Reviewer_ULHZ · 2023-11-22
>
> I thank the authors for their comprehensive edits, revisions, and clarifications. I have thoroughly looked through the authors’ revisions and their replies to my questions. I will adjust my rating to a 6, above the acceptance threshold.

---

> > ### Author Response · Authors · 2023-11-22
> >
> > Dear Reviewer ULHZ,
> >
> > Thank you for taking the time to review our responses. We are glad that our responses have addressed your comments. We really appreciate your feedback and acknowledgement. And thank you for increasing the score!

---

### Author Response · Authors · 2023-11-22
**General Response**

Thank you to all the reviewers for thoughtful and insightful feedback! We are pleased that reviewers are excited about the contributions of our work. We are encouraged that all reviewers appreciate that this paper presents a novel and crucial problem setup (Reviewers ULHZ, DnAy, CG4L, z7JT, Tnu1), find our method novel (Reviewers ULHZ, DnAy, CG4L, z7JT, Tnu1), and that empirical experiments demonstrate strong performance of FlexGen-EHR (Reviewers  ULHZ, DnAy, CG4L, z7JT, Tnu1). We thank the reviewers for the strong praise of our work and contributions.

We now highlight a few important points raised by reviewers that warrant inclusion in the general response:

1. **Ablation Study** [*Tnu1,DnAy*]: We have conducted an ablation study that demonstrates the superiority of the optimal transport component in FlexGen-EHR compared to traditional imputation methods like k-nearest neighbors (kNN).

2. **Partial Missing Temporal Features** [*ULHZ*]: Our experiments showcase the superior performance of FlexGen-EHR in scenarios where there are partial missing temporal features, highlighting its effectiveness.

3. **Clarifications**: We have taken the feedback into account and made clarifications in the paper regarding practical usage, computational complexity, and experimental setup/results to enhance clarity.

4. **Additional Discussion**: We have included a dedicated section in the paper that discusses limitations and outlines potential avenues for future research.

We thank all reviewers again for their thoughtful commentary. We worked hard to improve our paper, and we sincerely hope the reviewers find our responses informative and helpful. If you feel the responses have not sufficiently addressed your concerns to motivate increasing your score, we would love to hear what points of concern remain and how we can improve the work. Thank you again!

---

### Public Comment · ~Jun_Han6 · 2024-05-30
**How to ensure the temporal coherence of synthetic samples?**

Thank you for your great work!
FLEXGEN-EHR generates time-dependent feature of shape (T, d) by using a regular decoder. Say the time-dependent feature of shape (T, d) consists of x_1, x_2, ..., x_T, where each x_t has shape (1, d). How to ensure the samples x_t's at different time steps are coherent? Like in video generation, one challenging is to ensure the motion across different time steps is coherent.

---

> ### Public Comment · ~Huan_He2 · 2024-05-30
>
> Thank you for your great question. In our work, we take a different preprocessing way. We flatten temporal features into a 1-d vector ($T \times d$), where each value represents a quantity of interest at a time stamp. We agree the decoder step might not be generalizable to video generation.

---

### Meta-Review · Area_Chair_DXoR · 2023-12-06

**Metareview:**

This paper introduces FLEXGEN-EHR, a novel diffusion model, to generate realistic synthetic electronic health records (EHRs). The proposed method is capable of handling missing modalities in heterogeneous EHR data. The authors showed the method outperforms existing methods in fidelity and utility. It aligns diverse EHR features and is effective in privacy-sensitive settings where sharing original patient data is not feasible. The authors studied an important problem that is very useful in real-world situations with incomplete data. They propose a novel solution for the missing modality challenge by conceptualizing it as an optimal transport problem within the embedding space. I have thoroughly reviewed the paper and am deeply impressed by the significant contributions made by the authors. Their rebuttal to the reviewers' comments is not only adequate but also insightful, including their rebuttal to Reviewer CG4L who holds a different opinion. Given all these, I strongly advocate accepting this paper as a poster.

**Justification For Why Not Higher Score:**

Reviewers seem not excited enough to make a stronger case.

**Justification For Why Not Lower Score:**

The reviewers have many critical comments and the authors have addressed them adequately.

---

### Decision · Program_Chairs · 2024-01-16

Accept (poster)